# Inference of cell type-specific gene regulatory networks on cell lineages from single cell omic datasets

Shilu Zhang[1], Saptarshi Pyne[1], Stefan Pietrzak[1,2], Spencer Halberg[1,3], Sunnie Grace McCalla[1,4], Alireza Fotuhi Siahpirani[1,5], Rupa Sridharan[1,2] & Sushmita Roy ®[1,3] ✉

Cell type-specific gene expression patterns are outputs of transcriptional gene regulatory networks (GRNs) that connect transcription factors and signaling proteins to target genes. Single-cell technologies such as single cell RNA-sequencing (scRNA-seq) and single cell Assay for Transposase-Accessible Chromatin using sequencing (scATAC-seq), can examine cell-type specific gene regulation at unprecedented detail. However, current approaches to infer cell type-specific GRNs are limited in their ability to integrate scRNA-seq and scATAC-seq measurements and to model network dynamics on a cell lineage. To address this challenge, we have developed single-cell Multi-Task Network Inference (scMTNI), a multi-task learning framework to infer the GRN for each cell type on a lineage from scRNA-seq and scATAC-seq data. Using simulated and real datasets, we show that scMTNI is a broadly applicable framework for linear and branching lineages that accurately infers GRN dynamics and identifies key regulators of fate transitions for diverse processes such as cellular reprogramming and differentiation.

Transcriptional gene regulatory networks (GRNs) specify connections between regulatory proteins and target genes and determine the spatial and temporal expression patterns of genes[1,2]. These networks reconfigure during dynamic processes such as development or disease progression, to specify cell type specific expression levels. Recent advances in single cell omic techniques such as single cell RNA-sequencing (scRNA-seq) and single cell Assay for Transposase-Accessible Chromatin using sequencing (scATAC-seq)[3] enable collecting high resolution molecular phenotypes of a developing system and offer unprecedented opportunities for the discovery of cell type-specific regulatory networks and their dynamics. However, computational methods to systematically leverage these datasets to identify regulatory networks driving cell type-specific expression patterns are limited.

Existing methods of network inference from single cell omic data[4–16] have primarily used transcriptomic measurements and have low recovery of experimentally verified interactions[17,18]. Recently a small number of methods have attempted to integrate scRNA-seq and scATAC-seq datasets[19–21] to examine gene regulation, however, many of these methods focus on defining cell clusters and the network is defined entirely based on accessible sequence-specific motif matches. This restricts the class of regulators that can be incorporated into the regulatory network to those with known motifs. Furthermore, existing methods infer a single GRN for the entire dataset or do not model the cell population structure which is important to discern dynamics and transitions in the inferred networks for cell type-specificity. To overcome the limitations of existing methods, we have developed single-cell Multi-Task Network Inference (scMTNI), a multi-task learning

[1]Wisconsin Institute for Discovery, University of Wisconsin-Madison, Madison, WI, USA. [2]Department of Cell and Regenerative Biology, University of Wisconsin-Madison, Madison, WI, USA. [3]Department of Biostatistics and Medical Informatics, University of Wisconsin-Madison, Madison, WI, USA. [4]Laboratory of Genetics, University of Wisconsin-Madison, Madison, WI 53706, USA. [5]Present address: Department of Bioinformatics, Institute of Biochemistry and Biophysics, University of Tehran, Tehran, Iran. ✉e-mail: sroy@biostat.wisc.edu

framework that integrates the cell lineage structure, scRNA-seq and scATAC-seq measurements to enable joint inference of cell type-specific GRNs. scMTNI takes as input a cell lineage tree, scRNA-seq data and scATAC-seq based prior networks for each cell type. scMTNI uses a probabilistic prior to incorporate the lineage structure during network inference and outputs GRNs for each cell type on a cell lineage. We performed a comprehensive benchmarking study of multi-task learning approaches including scMTNI on simulated data and show that incorporation of multi-task learning and tree structure is beneficial for GRN inference.

We applied scMTNI to a previously unpublished scRNA-seq and scATAC-seq time course dataset for cellular reprogramming in mouse and two published scRNA-seq and scATAC-seq cell-type specific datasets for human hematopoietic differentiation. We demonstrate the advantage of scMTNI's framework to integrate scATAC-seq and scRNA-seq datasets for inferring cell type specific GRNs on linear and branching lineage topologies. We examined how the inferred networks change along the trajectory and identified regulators and network components specific to different parts of the lineage tree. Our predictions include known as well as previously uncharacterized regulators of cell populations transitioning to different lineage paths, providing insight into regulatory mechanisms associated with reprogramming efficiency and hematopoietic specification.

## Results

### Single-cell Multi-Task learning Network Inference (scMTNI) for defining regulatory networks on cell lineages

We developed scMTNI, a multi-task graph learning framework for inferring cell type-specific gene regulatory networks from scRNA-seq and scATAC-seq datasets (Fig. 1a), where a cell type is defined by a cluster of cells with a distinct transcriptional, and, if available, accessibility profile. scMTNI models a GRN as a Dependency network[22], a probabilistic graphical model with random variables representing genes and regulators, such as transcription factors (TFs) and signaling proteins.

scMTNI takes as input cell clusters with gene expression and accessibility profiles and a lineage structure linking the cell clusters (Fig. 1). Such inputs can be obtained from existing methods for integrative clustering[23] and lineage construction[24]. scMTNI uses the scATAC-seq data for each cell cluster to define cell type-specific sequence motif-based TF-target interactions (e.g., a motif for a particular TF, which is accessible only in specific cell types will result in a TF-target interaction only in those cell types) which are used as a prior to guide network inference (Methods). scMTNI can also take bulk ATAC-seq data for corresponding cell types to generate cell type-specific prior networks or cell type-agnostic priors derived from sequence-specific motifs that in turn could be filtered with relevant ATAC-seq data. scMTNI's multi-task learning framework incorporates a probabilistic lineage tree prior, which uses the lineage tree structure to influence the similarity of gene regulatory networks on the lineage. This lineage tree prior models the change of a GRN from a start state (e.g., progenitor cell state) to an end state (e.g., more differentiated state) as a series of individual edge-level probabilistic transitions. The output of scMTNI is a set of cell type-specific GRNs one for each cell cluster in the lineage tree. scMTNI is able to incorporate both linear lineage and tree-based lineage structure. scMTNI takes known cell lineage tree structure or computationally inferred cell lineage using, for example, a minimum spanning tree (MST[24]) approach on scRNA-seq data. While scMTNI was developed to incorporate both scRNA-seq and scATAC-seq data, it can be applied to situations where scATAC-seq, and therefore a cell type-specific prior network, is not available. We refer to the versions of our approach as scMTNI+Prior and scMTNI depending upon whether it uses prior knowledge or not. The output networks of scMTNI are analyzed using two dynamic network analysis methods: edge-based k-means clustering and topic models (Fig. 1b).

These approaches identify key regulators and subnetworks associated with a particular cell cluster or a set of cell clusters on a branch.

### Multi-task learning algorithms outperform single-task algorithms for single cell network inference

To evaluate scMTNI and other existing algorithms with known ground truth networks on single-cell transcriptomic data, we set up a simulation framework, which entailed creation of a cell lineage, generating synthetic networks and corresponding single-cell expression datasets for each cell type on the lineage (Fig. 2a). We used a probabilistic process of network structure evolution to generate the network structure for three cell types, each containing 15 regulators and 65 genes and between 202–239 edges (Methods). Next, we applied BoolODE[17] to simulate the in silico single-cell expression data using each cell type's generated network. To mimic the sparsity in single-cell expression data, we set 80% of the values to 0. We created three datasets with different numbers of cells: 2000, 1000, and 200, referred here as datasets 1, 2, and 3.

We asked whether multi-task learning is beneficial compared to single-task learning for network inference from scRNA-seq data. To this end, we compared scMTNI and four other multi-task learning algorithms, MRTLE[25], GNAT[26], Ontogenet[27], and AMuSR[28] to three single-task algorithms, LASSO regression[29], INDEP, and SCENIC[30] (Methods). Of these methods, only SCENIC uses a non-linear regression model while the others are based on linear models. INDEP is similar to scMTNI but does not incorporate the lineage prior. Each algorithm was applied within a stability selection framework and evaluated with Area under the Precision recall curve (AUPR) and F-score of top $k$ edges, where $k$ is the number of edges in the true network (Fig. 2b, c). On dataset 1, based on AUPR, scMTNI, MRTLE, and AMuSR are able to recover the network structure better than the other multi-task learning and single-task learning algorithms (Fig. 2b). Ontogenet performs better than the single-task learning algorithms in at least two cell types. Finally, GNAT performs comparably to the single-task learning algorithms. When comparing algorithms based on F-score of top $k$ edges, we have similar observations that scMTNI and MRTLE have a better performance than other algorithms (Fig. 2c). Ontogenet performs better than LASSO and INDEP in at least two cell types, and comparable to SCENIC, except that Ontogenet in cell type 3 is worse than SCENIC. GNAT is comparable to the single-task learning algorithms for at least 2 of the cell types. The low F-score of AMuSR is because the inferred networks are too sparse, with fewer than 100 edges, while the other algorithms inferred similar number of edges as the true networks. These results remain consistent for datasets 2 and 3 which have fewer cells (1000 and 200, respectively); scMTNI and MRTLE remain superior in performance than other algorithms measured by both AUPR and F-score (Fig. 2b, c). We expect scMTNI to be better since the network simulation procedure is similar, but the data generation process is different and independent from scMTNI's model. Finally, we aggregated the results across all three cell types and datasets to obtain an overall comparison of the algorithms. Here we considered algorithms across all parameter settings tested as well as the best parameter setting determined by the best F-score or AUPR. Based on the AUPR of "all parameter setting", we found that multi-task learning methods, especially scMTNI and MRTLE are generally better than single-task learning methods with higher AUPRs (Supplementary Fig. 1A, C). AMuSR also outperformed the single-task algorithms based on AUPRs, although this was not as significant as MRTLE and scMTNI. When considering the "best parameter setting", the methods were not significantly different when using AUPR, though MRTLE and scMTNI had the highest AUPR (Supplementary Fig. 1B, D). When using the F-score, scMTNI and MRTLE remained top performing algorithms for the "all parameter setting" (Supplementary Fig. 2A, C) and the "best parameter setting" (Supplementary Fig. 2B, D). Further, GNAT and Ontogenet had a higher F-score than the single-task learning method

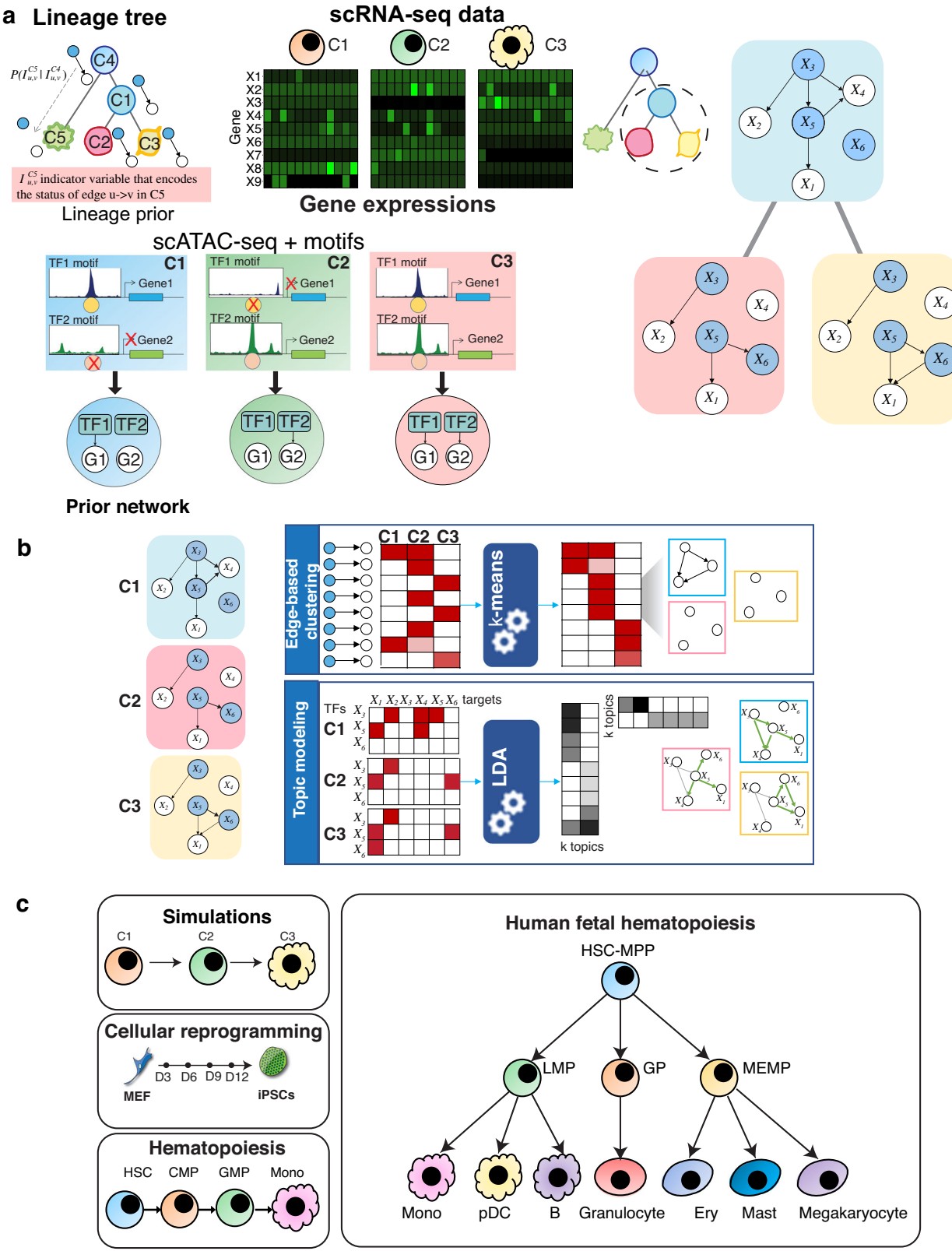

LASSO for the "all parameter" and "best parameter" settings, respectively. AMuSR suffered on the F-score metric due to the high sparsity in the inferred networks. Across different single-task algorithms, LASSO had the worst performance. Overall, the results on the simulated networks suggest that multi-task learning algorithms have a better performance than single-task algorithms for network inference on sparse datasets such as single-cell transcriptomic data. Furthermore, scMTNI and MRTLE are able to more accurately infer networks than other multi-task learning algorithms.

### Inference of gene regulatory networks of somatic cell reprogramming to induced pluripotent stem cells

Cellular reprogramming is the process of converting cells in a differentiated state to a pluripotent state and is important in regenerative

**Fig. 1 | An overview of the scMTNI framework. a** scMTNI takes as input a cell lineage tree and cell type-specific scRNA-seq data and cell type-specific prior networks derived from scATAC-seq datasets. If scATAC-seq data is not available, bulk or sequence-based prior networks can be used for the cell types. The output of scMTNI is a set of cell type-specific gene regulatory networks for each cell type on the cell lineage tree. **b** The output networks of scMTNI are analyzed using two dynamic network analysis methods: edge-based k-means clustering and Latent Dirichlet Allocation (LDA) based topic models to identify key regulators and subnetworks associated with a particular cell cluster or a set of clusters on a branch. **c** Datasets used with scMTNI. The simulation data comprised a linear trajectory of three cell types, while the three real datasets came from a reprogramming time-series process, immunophenotypic cell types identified during human adult hematopoietic differentiation, and immunophenotypic blood cells during human fetal hematopoiesis. MEF mouse embryonic fibroblast, iPSCs induced pluripotent cells, HSC hematopoietic stem cell, CMP common myeloid progenitor, GMP granulocyte-macrophage progenitors, Mono monocyte, HSC-MPP hematopoietic stem cells and multipotent progenitors, LMP lymphoid-myeloid progenitors, MEMP MK-erythroid-mast progenitors combined with cycling MEMPs, GP granulocytic progenitors, Ery erythroid cells, pDC plasmacytoid dendritic cells.

medicine as well as for generating patient-specific disease models. However, this process is inefficient as a small fraction of cells get reprogrammed to the pluripotent state[31]. To gain insight into gene regulatory networks that govern the dynamics of this process, we profiled single cell accessibility (scATAC-seq) during reprogramming of mouse embryonic fibroblasts (MEFs) to the induced pluripotent state and four intermediate timepoints, day 3, day 6, day 9, and day 12, to constitute a dataset of 6 timepoints. We used LIGER to integrate the scRNA-seq and scATAC-seq datasets (Fig. 3a, b) and identified 8 clusters (Methods). Of these clusters, C4 is MEF-specific while C5 is ESC-specific (Fig. 3c, d) and showed good integration of the scRNA-seq and scATAC-seq profiles (Supplementary Fig. 3). We removed C6 as it did not have scRNA-seq cells and applied a minimum spanning tree (MST[24]) approach to construct the cell lineage tree from the 7 cell clusters with both scRNA-seq and scATAC-seq (Methods, Fig. 3e). The MEF-specific cluster (C4) is at one end of the tree, while the ESC-specific cluster (C5) is at the other end. This is consistent with the starting and end state of the reprogramming process and we considered C4 to represent the root of the tree. The other clusters represented a mix of cells from different time points, which is consistent with the level of heterogeneity of the reprogramming system[32]. We further verified the identity of these intermediate clusters with a Monocle based trajectory analysis[33] which shows that C7, C2, and C3 represent cells that might exit the trajectory towards reprogramming and C8 represents cells upstream of this point (Supplementary Fig. 4).

We applied scMTNI, scMTNI+Prior (scMTNI with prior network), INDEP, INDEP+Prior (INDEP with prior network), SCENIC and additionally CellOracle to this dataset (Fig. 3f). We included CellOracle as it combines scRNA-seq and scATAC-seq data, by using accessibility to restrict the set of edges selected based on expression. We used the matched scATAC-seq clusters to obtain TF-target prior interactions for each scRNA-seq cluster needed for INDEP+Prior, scMTNI+Prior and CellOracle (Methods). We assessed the quality of the inferred networks by comparing to multiple gold standard datasets in mouse embryonic stem cells (mESCs, Table 1): one derived from ChIP-seq experiments ("ChIP") from ESCAPE or ENCODE databases[34,35], one from regulator perturbation experiments ("Perturb")[34,36], and the third from the intersection of edges in ChIP and Perturb ("ChIP + Perturb"). We first compared the performance of the methods using F-score on the top 500, 1k, and 2k edges across methods (Fig. 3f, Supplementary Figs. 5, 6). On Perturb, CellOracle and scMTNI+Prior had the best performance, beating other algorithms significantly. On ChIP, SCENIC and CellOracle were the best performing methods. Finally, on Perturb + ChIP, CellOracle and scMTNI+Prior had the best performance. Although CellOracle had high F-scores, its inferred GRNs included a substantially smaller number of regulators (7–11) compared to SCENIC or scMTNI + Prior (29–36). In addition to F-score, we also considered the number of predictable TFs as an additional metric (Supplementary Fig. 7, Methods). This is defined as the number of individual TFs whose targets had a significant overlap with the gold standard. Higher the number of predictable TFs, the better is a method. On ChIP, scMTNI + Prior had the highest average number of predictable TFs. scMTNI had the highest number of predictable TFs for the Perturb, Perturb + ChIP datasets followed closely by scMTNI + Prior. Overall,

scMTNI+Prior had among the highest F-scores, high number of predictable TFs and a greater coverage of the gold standards compared to competing methods using expression alone (SCENIC) as well as those that either incorporated accessibility information (CellOracle, INDEP + Prior) or cell lineage information (scMTNI).

To perform an initial assessment of the network dynamics on the cell lineage, we computed F-score between each pair of inferred networks defined by the top 4k edges (Fig. 3g). Both scMTNI and scMTNI + Prior networks diverged in a manner consistent with the lineage structure. scMTNI networks formed three groups of cell types, (C4, C8, C1, C7), (C2, C3) and (C5 (ESC)). scMNTI + Prior found similar groupings but placed C5 (ESC) closer to (C1, C7, C8, C4) branch. Both methods showed that C5 is closest to C1, which could be an important transitioning state of cells during reprogramming. SCENIC showed similarity among C1, C4, C7, however had lower similarity scores for most pairwise comparisons which made it difficult to discern a clear lineage structure. CellOracle topology identified the (C2, C3) group, but placed it under a subtree with (C4, C8), which, though feasible given the heterogeneity of the system, is less consistent with the gradual progression of the reprogramming process through the intermediate C7 state. The networks inferred by the other methods were very dissimilar which is biologically unrealistic given the high heterogeneity of the reprogramming system with several intermediate populations[32]. Overall, these results suggest that scMTNI+Prior recovered regulatory networks of high quality and the networks exhibit a gradual rewiring of structure from the MEF to the pluripotent state.

## scMTNI predicts key regulatory nodes and GRN components that are rewired during reprogramming

To gain insight into the regulatory mechanisms of cell populations that successfully reprogram versus those that do not and to further characterize these different cell clusters, we examined the rewired network components in each cell type-specific network inferred by scMTNI + Prior. We used two complementary approaches: k-means edge clustering and Latent Dirichlet Allocation (LDA, Methods). In the k-means edge clustering approach, we represented each edge in the top 4k confidence set of any cell cluster, by a vector of confidence scores in each cell cluster-specific network (if an edge is not inferred in the network it is assigned a weight of 0). Next, we clustered edges based on their edge confidence pattern into 20 clusters determined by the Silhouette Index coefficient optimization (Fig. 4a). The largest "edge clusters" exhibited interactions specific to one cell cluster (e.g., E4, E6, E7, E11, E13, E15, and E16), while smaller clusters exhibited conserved edges for more than one cell cluster (e.g., E2, E5, E12). To interpret these edge clusters, we identified the top regulators associated with each of the edge clusters (Fig. 4b). E16, which was MEF-specific (C4) had *Npm1, Nme2, Thy1, Ddx5,* and *Loxl2* as the top regulators which are known MEF-specific genes. In contrast, E11, which was ESC-specific (C5) had *Klf4, Sp1, Sp3* as some of its top regulators, which have known roles in stem cell maintenance (*Klf4*), or are essential for early development (*Sp1*[37]) and post natal development (*Sp3*[38]). Edge clusters that shared edges across multiple cell clusters, e.g., E5 (C4, C8, and C1), shared some of the top-ranking regulators such as *Npm1* and *Thy1* with the

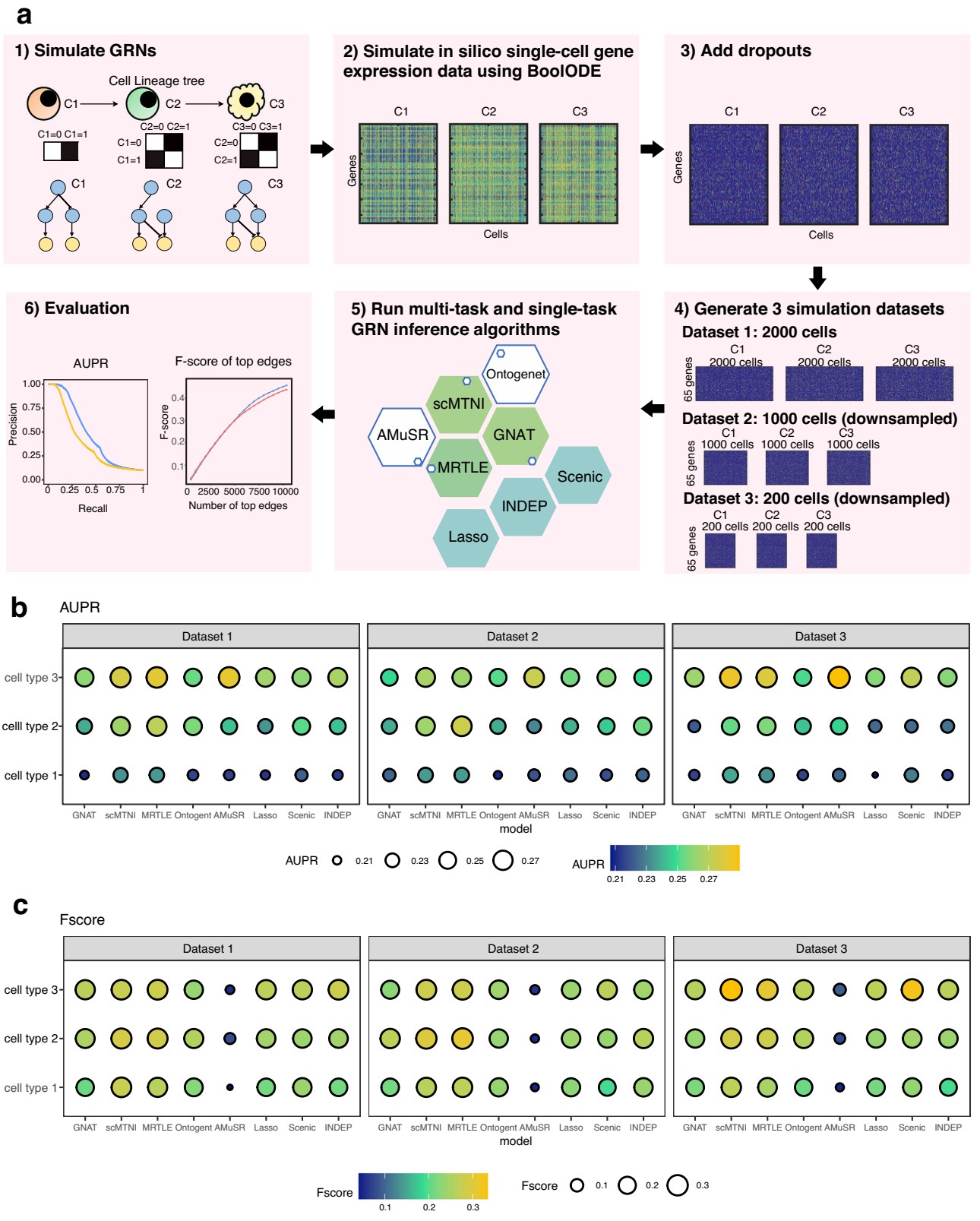

**Fig. 2 | Benchmarking algorithms on simulated data. a** Simulation framework for scMTNI. We first simulate GRNs for cell types across a cell lineage tree. Next, we generate in silico single-cell gene expression data for each cell type using BoolODE using the simulated GRNs and add 80% zeros in the simulation data. Then, we apply five multi-task learning algorithms and three single-task learning algorithms for GRN inference to the simulated datasets and predict networks in stability selection framework. We compare the performance of these algorithms based on area under precision and recall curve (AUPR) and F-score of top edges. **b** AUPR comparing inferred networks to ground truth networks of simulated datasets 1, 2, 3. **c** F-score comparing top $K$ edges in the inferred networks to those in the ground truth networks of simulated datasets 1, 2, 3 (cell type 1: $K = 202$, cell type 2: $K = 217$, cell type 3: $K = 239$). The brighter and larger the circle the better the performance of the algorithm. Source data are provided as a Source Data file.

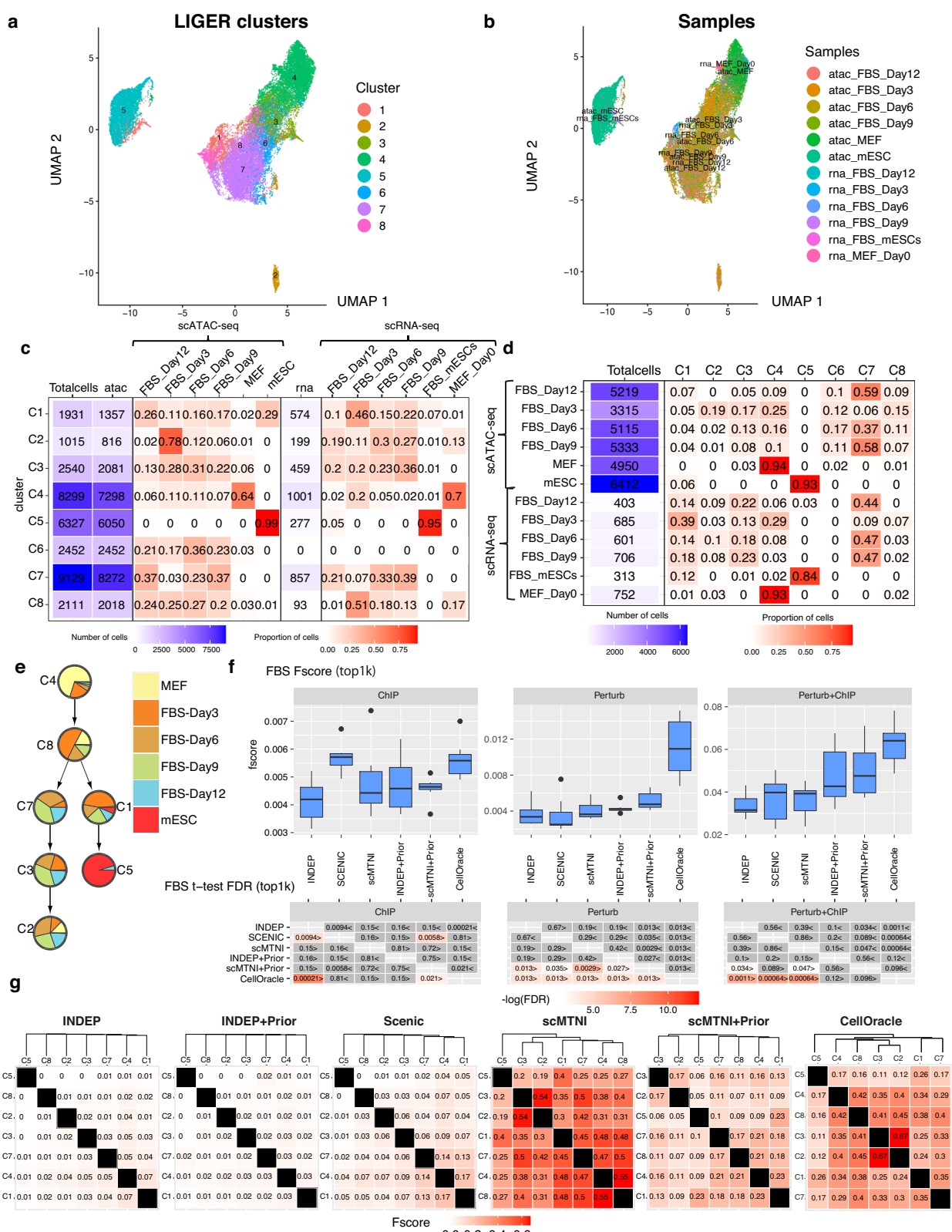

MEF-specific cluster and also identified other fibroblast-specific genes such as *Col5a2* and *Ybx1*. Finally, E2 which comprised shared edges between cell clusters C1 and C5, contained *Esrrb*, as its top regulator (Fig. 4b). *Esrrb* plays an important role for establishing and maintaining the pluripotency network[39]. This further supports the lineage structure that C1 likely represents a population of cells that are committed to becoming pluripotent.

While the k-means analysis identified regulatory hubs specific to individual cell clusters, it was challenging to identify entire sub-networks that rewired at specific branch points because it treats each edge independently. We developed an approach by adopting Latent Dirichlet Allocation (LDA) that was recently used to study regulatory network rewiring from transcription factor ChIP-seq datasets[40] (Methods). In this approach, each TF is treated as a "document" and

**Fig. 3 | Inference of cell-type specific networks of mouse cellular reprogramming data. a** UMAP of LIGER cell clusters on the scATAC-seq data and scRNA-seq data. **b** UMAP depicting the sample labels of the scATAC-seq and scRNA-seq data from mouse cellular reprogramming. **c** The distribution of samples in each LIGER cluster. **d** The distribution of LIGER clusters in each sample. **e** Inferred lineage structure for scMTNI linking the 7 cell clusters with scRNA-seq measurements. **f** F-score of top 1k edges in predicted networks of scMTNI, scMTNI+Prior, INDEP, INDEP+Prior, SCENIC and CellOracle compared to three gold standard datasets: ChIP, Perturb and Perturb+ChIP. The top boxplots show the F-scores of $n = 7$ cell clusters, while the bottom heatmaps show FDR corrected $t$-test comparing the F-scores of the row algorithm to that of the column algorithm. The two-sided paired $t$-test is conducted on F-scores of $n = 7$ cell clusters for every pair of algorithms. A

FDR < 0.05 was considered significantly better. The sign < or > specifies whether the row algorithm's F-scores were worse or better than the column algorithm's F-scores. The color scale is specified by $-\log(FDR)$, with the red color proportional to significance. Non-significance is colored in gray. In the boxplot, the horizontal middle line of each plot is the median. The bounds of the box are 0.25 quantile ($Q_1$) and 0.75 quantile ($Q_3$). The upper whisker is the minimum of the maximum value and $Q_3 + 1.5*IQR$, where $IQR = Q_3 - Q_1$. The lower whisker is the maximum of the minimum value and $Q_1 - 1.5*IQR$. **g** Pairwise similarity of networks from each cell cluster using F-score on the top 4k edges. Rows and columns are ordered based on the dendrogram created using the F-score similarity. Source data are provided as a Source Data file.

target genes are treated as "words" in the document. Each document (TF) is assumed to have words (genes) from a mixture of topics, each topic in turn interpreted as a pathway. TFs across cell clusters are treated as separate documents. We applied LDA with $k = 10$ topics (Fig. 4c, d, Supplementary Figs. 8–10), and examined each of the topics based on their Gene Ontology process enrichment (Supplementary Fig. 11), and the tendency and identity of specific regulators to rewire across the cell clusters. Topics 3 and 6 are enriched for cell cycle terms (Supplementary Fig. 11). Other processes associated with these topics included immune response (topic 1), developmental processes (topics 1, 3 and 8), electron transport (topic 9), and chromosome organization (topic 10). Topic 3 networks were among the most divergent networks across the cell populations and identified several known regulators of pluripotency (Fig. 4c). In particular, *Esrrb* was a hub in C5 (ESC) and C1 (closest to ESC) but absent in the other cell clusters.

We used the LDA analysis to further characterize cell populations that become pluripotent (C1-C5 branch), and those that remain stalled (C7-C3-C2 branch) by identifying regulators that gained or lost connections between these two branches. Several topics included

regulators that showed a difference in connectivity between these branches including topics 2, 3, 4, 6, 8, and 9. The regulators that gained edges in the pluripotency branch compared to the stalled branch included cell cycle regulators (*Top2a, Ccnb1*: topic 3) and known pluripotency genes (*Esrrb*: topic 3 and *Klf4*: topic 4, Fig. 4d). In contrast, regulators that gained connections in C7-C3-C2 branch relative to the C1-C5 branch (or maintained connections similar to C4), included MEF-specific genes such as *Loxl2, Fosl2* (topic 2), *Aebp1* (topic 6), *Hoxd13* (topic 8), and *Fosl1, Nme2* and *Ccng1* (topic 9). *Nme2* is known to regulate *Myc*, which is one of the four reprogramming factors[41]. *Aebp1*, associated with fibroblast differentiation[42], and *Loxl2*, associated with connective tissue[43,44], persisted in all three cell clusters in the stalled branch (C7-C3-C2). Overall, our analysis indicated that in cell populations that do not reprogram successfully, cell cycle regulators have lower connectivity while several of the MEF regulators (e.g., *Nme2, Aebp1*) persist or gain connections. These new predicted regulators can be perturbed to examine the impact on cellular reprogramming efficiency.

### Inferring gene regulatory networks in human hematopoietic differentiation

To examine the utility of scMTNI in a different cell fate specification system, we applied scMTNI to a published scATAC-seq and scRNA-seq dataset for human adult hematopoietic differentiation[45]. This dataset profiled accessibility and transcriptomic state of immunophenotypic populations that were sorted based on cell surface markers and enabled studies of how multipotent progenitors transition into lineage-specific cell states. We considered the cell populations profiled with both scATAC-seq and scRNA-seq datasets: hematopoietic stem cell (HSC), common myeloid progenitor (CMP), granulocyte-macrophage progenitors (GMP) and monocyte (Mono). These populations are known to be heterogeneous comprising multiple sub-populations[45]. To identify these sub-populations, we again applied LIGER[23] and identified 10 integrated clusters of RNA and accessibility (Fig. 5a–d). Most clusters exhibited a mixed composition: C8 is mainly composed of HSCs but also included CMP0 cells; C6 and C9 are composed of GMP and CMP0 cells. C1 (73 cells) and C4 (37 cells) were mainly composed of Mono cells and were combined into C1. C5 had too few RNA cells (22 cells) and was excluded from further analysis. We next inferred a cell lineage tree from these 8 cell clusters using a minimal spanning tree approach[24] as described in the reprogramming study (Fig. 5e, Methods). As C8 is largely made up of HSC cells and HSC is the starting cell type, we treated C8 as the root of the lineage.

We applied the same set of network inference algorithms to this dataset as the reprogramming dataset: scMTNI, scMTNI+Prior, INDEP, INDEP+Prior, SCENIC and CellOracle. We assessed the quality of the inferred networks from each method by comparing them to gold-standard edges from published ChIP-seq and regulator perturbation assays from several human hematopoietic cell types. This included ChIP-seq datasets from the UniBind database (Unibind[46]), ChIP-seq (Cus_ChIP) and regulator perturbation (Cus_KO) experiments in the GM12878 lymphoblastoid cell line from Cusanovich et al.[47] and the

**Table 1 | The statistics of the gold standard datasets used for the mouse reprogramming and human hematopoiesis studies**

| Dataset | Gold standards | Number of TFs | Number of targets |
|---|---|---|---|
| Mouse reprogramming | ChIP | 54 | 31,367 |
| | Perturb | 179 | 21,019 |
| | Perturb + ChIP | 47 | 6109 |
| Human hematopoiesis | Hematopoietic stem cells (HSC) | 6 | 9173 |
| | CD14_monocytes | 1 | 6523 |
| | megakaryocytes | 4 | 8733 |
| | erythroid_progenitors | 1 | 7955 |
| | R3R4_erythroid_cells | 1 | 8494 |
| | macrophages | 1 | 163 |
| | CD34_hematopoietic_stem_cells-derived_proerythroblasts | 3 | 5847 |
| | T-cells | 3 | 6189 |
| | B-cells | 1 | 7036 |
| | GM_B-cells | 48 | 10,597 |
| Human hematopoiesis | UniBind | 56 | 10,621 |
| | Cus_ChIP | 149 | 6179 |
| | Cus_KO | 50 | 6108 |
| | Cus_KO + Cus_ChIP | 26 | 2124 |
| | Cus_KO + UniBind | 12 | 2020 |

For mouse reprogramming, shown are network statistics for the mouse embryonic stem cell (ESC) line from ESCAPE[34] and ENCODE[35] databases and Nishiyama et al.[36]. For the human hematopoietic studies, shown are network statistics for the gold standard datasets obtained from the UniBind database[46] and Cusanovich et al.[47].

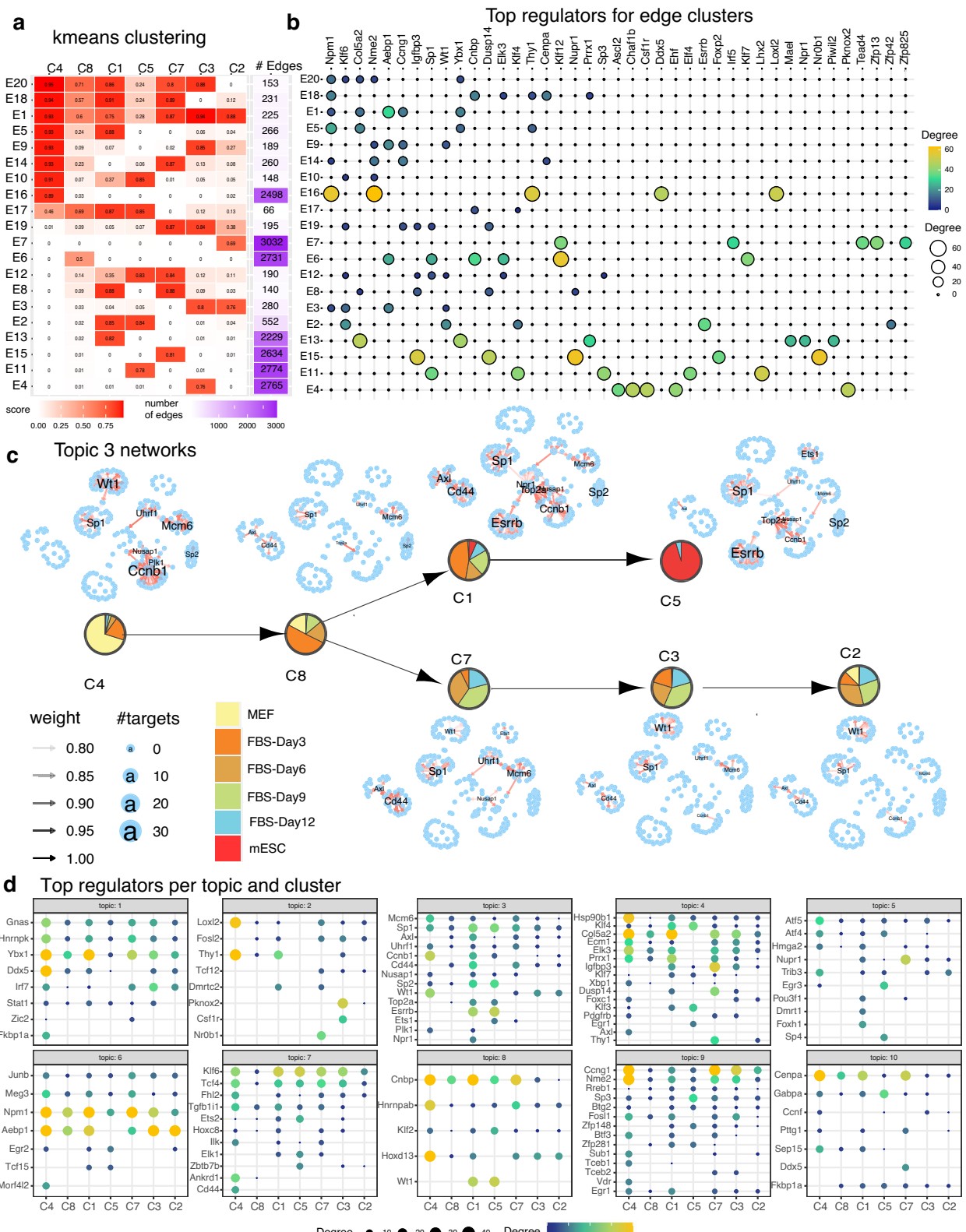

**Fig. 4 | Network dynamics analysis of GRNs from cellular reprogramming.**
**a** k-means clustering analysis of top 4k edges in inferred networks. Shown are the mean profiles of edge confidence of 20 edge clusters. Each row corresponds to an edge cluster and each column corresponds to a cell cluster. The red intensity corresponds to the average confidence of edges in that cluster. Shown also are the number of edges in the edge cluster. **b** Top 5 regulators for each edge cluster. Shown are only regulators that have at least 10 targets in any edge cluster. The size and brightness of the circle is proportional to the number of targets. **c** LDA topic 3 networks along the cell lineage. The layout of each network is the same, edges present in a particular cell cluster are shown in red. Labeled nodes correspond to regulators with degree larger than 10. **d** Top cell cluster-specific regulators for each topic. Shown are only regulators that have at least 10 targets in any cell cluster. The more yellow and larger the circle, the greater are the number of targets for the regulator. Source data are provided as a Source Data file.

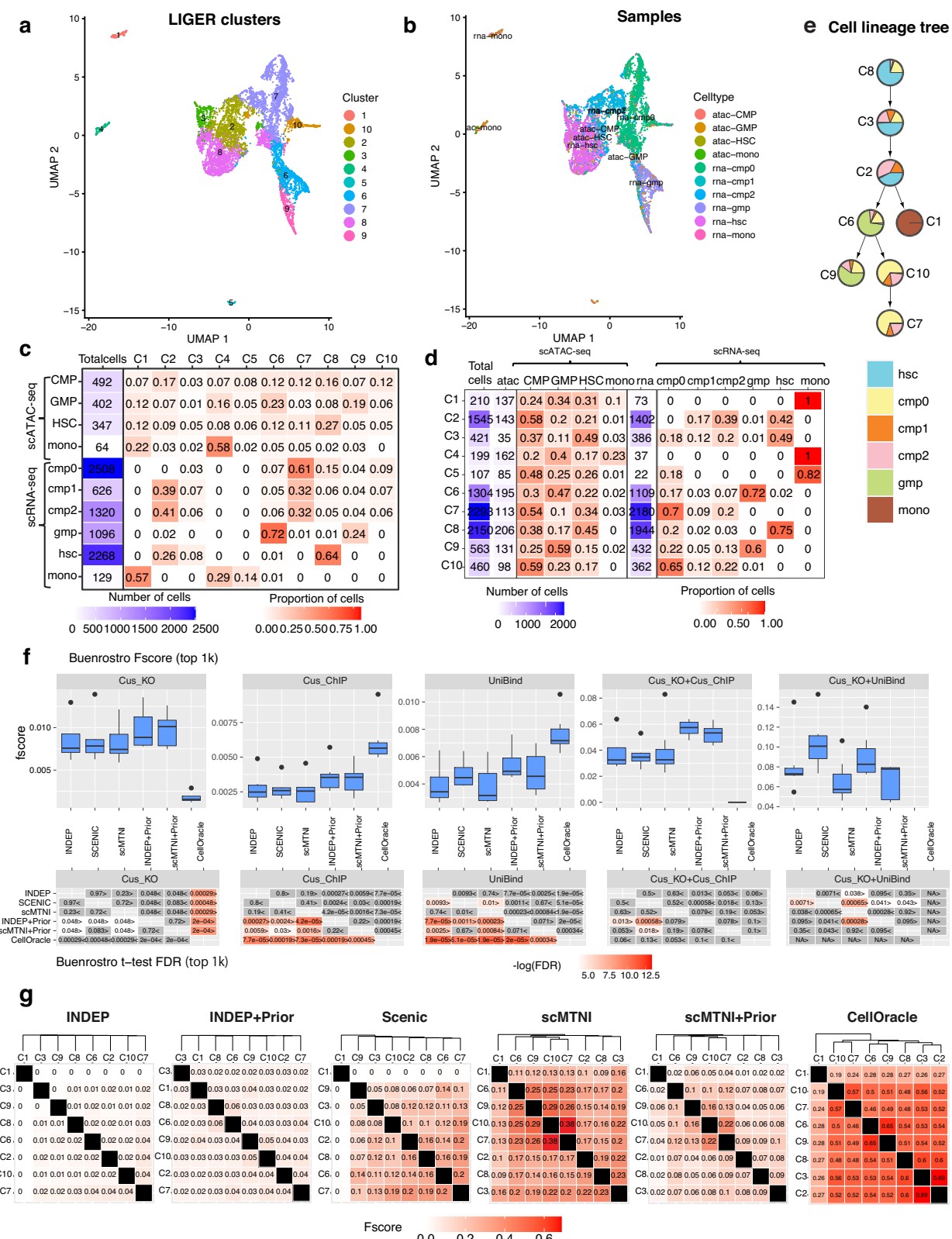

intersection of ChIP and perturbation studies (Cus_KO+Cus_ChIP, Cus_KO+Unibind). In total, we had five gold standard networks. We used F-score and the number of predictable TFs of the top 500, 1k, 2k edges in the inferred network (Methods, Fig. 5f, Supplementary Fig. 12). The relative performance of the algorithms depended upon the gold standard. Algorithms that did not use priors (INDEP, SCENIC and scMTNI) performed comparably (with no significant difference)

on three of the five gold standards. On Unibind and Cus_KO+Unibind, SCENIC is significantly better than INDEP and scMTNI (Fig. 5f, Supplementary Fig. 13). Methods that used prior knowledge, CellOracle, INDEP+Prior, scMTNI+Prior, were generally better than methods without priors for the ChIP-based datasets (Cus_ChIP, Unibind). CellOracle performs better than INDEP+Prior and scMTNI+Prior on Cus_ChIP and Unibind, but is outperformed by all methods on any of

**Fig. 5 | Inference of cell type-specific networks for human hematopoietic differentiation data. a** UMAP of LIGER cell clusters of the scATAC-seq and scRNA-seq data. **b** UMAP depicting the original cell types (samples) with scATAC-seq and scRNA-seq data. **c** The distribution of cell clusters in each sample. **d** The distribution of samples in each LIGER cluster. **e** Inferred lineage structure linking the eight cell clusters with scRNA-seq data. **f** Boxplots showing F-scores of $n = 7$ cell clusters (all cell clusters excluding C1) for top 1k edges in predicted networks from scMTNI, scMTNI+Prior, INDEP, INDEP+Prior, SCENIC and CellOracle compared to gold standard datasets (top). FDR-corrected $t$-test to compare the F-score of the row algorithm to the F-score of the column algorithm (bottom). The two-sided paired $t$-test is conducted on F-scores of $n = 7$ cell clusters for every pair of algorithms. A

FDR < 0.05 was considered significantly better. The sign < or > specifies whether the row algorithm's F-scores were worse or better than the column algorithm's F-scores. The color scale is specified by $-\log(\text{FDR})$, with the red color proportional to significance. Non-significance is colored in gray. In the boxplot, the horizontal middle line of each plot is the median. The bounds of the box are 0.25 quantile ($Q_1$) and 0.75 quantile ($Q_3$). The upper whisker is the minimum of the maximum value and $Q_3 + 1.5 \cdot IQR$, where $IQR = Q_3 - Q_1$. The lower whisker is the maximum of the minimum value and $Q_1 - 1.5 \cdot IQR$. **g** Pairwise similarity of networks from each cell cluster using F-score on the top 5k edges. Rows and columns ordered by hierarchical clustering using F-score as the similarity measure. Source data are provided as a Source Data file.

the regulator perturbation datasets. INDEP+Prior and scMTNI+Prior are comparable across the gold standard datasets with no significant difference in performance (Fig. 5f, Supplementary Fig. 13). Based on number of predictable TFs in the predicted networks (Supplementary Fig. 14), INDEP+Prior and scMTNI+Prior recovered more predictable TFs especially in KO experiments, while CellOracle recovered more predictable TFs in Cus_ChIP and UniBind. For the Unibind dataset, we had ChIP-seq based gold standard edges for different blood cell types, with 1 to 48 transcription factors (Table 1). Of the 10 cell types, methods that used priors performed significantly better than methods that did not on the GM_B-cells and Hematopoietic Stem Cells (HSCs) which had the largest number of TFs (Supplementary Figs. 15, 16). However, CellOracle had much lower performance in other cell types and was outperformed by methods with and without priors, likely because of the smaller number of TFs in these datasets. The number of predictable TFs per dataset and method was generally low with the exception of GM_B-cells where methods with priors were better than methods without priors (Supplementary Fig. 17). However, these gold standards were much smaller and therefore can assess smaller portion of the inferred networks.

We next examined the inferred networks for the extent of change on the lineage structure (Fig. 5g). The single-task learning methods INDEP and INDEP+Prior exhibited a low overlap across each pair of cell lines and did not as such obey the lineage structure. SCENIC recovers part of the lineage structure, but placed C7 (common myeloid) close to C6 (granulocyte-macrophage progenitors (GMP)) rather than C10, which has similar sample composition as C7. In contrast, scMTNI and scMTNI+Prior were able to find two groups of cell types, one corresponding to the HSC and CMP2 branch consisting of C8, C3, and C2, and the second corresponding to the CMP0, CMP1, and GMP branch (C6, C9, C10, and C7). CellOracle also inferred a similar tree with small variations within these two groups. For this dataset, the addition of accessibility or lineage information was helpful to capture realistic extents of network level changes.

### Inferring shared and lineage-specific regulators for hematopoietic differentiation

Similar to our cellular reprogramming study, we examined the scMTNI +Prior networks to identify cell type-specific regulators and network components (Fig. 6) with k-means and LDA analysis. We applied k-means edge clustering to the union of top 5k edges in any of the cell clusters and identified 19 edge clusters (Methods). Compared to the reprogramming study, a larger portion (94% vs 86%) of the edges are specific to one cell cluster (Fig. 6a). We used these edge clusters to examine the differences and similarities at the branch between the CMP clusters (C7, C10), and the GMP clusters (C6 and C9). Edge cluster E12 was specific to C7 and C10, E18 was specific to C6 and C9, and E19 shared edges from C6, C9, C10, C7. Both E19 and E12 had *YBX1* and *TSC22D3* as top regulators (Fig. 6b). *YBX1* is known to direct fate of HSCs with high expression in myeloid progenitor cells[48] and involved in monocyte/macrophage differentiation[49]. *TSC22D3*, which is a glucocorticoid leucine zipper[50], is involved in differentiation of

hematopoietic stem cells[51]. E12 additionally had *KLF1, FLI1, S100A4* as top regulators. *KLF1* is an essential regulator for the erythroid lineage[52,53], which is derived from the myeloid progenitor cells. *FLI1* also plays a role in erythroid lineage by regulating the Erythpoetin protein[54], suggesting these cells are committed to the erythroid lineage. In contrast, E18 which shared edges between C6 and C9 identified immune system-related regulators such as *IRF8* and *NFKBIA* which have been associated with general lymphoid development (*IRF8*[55]) or specific lineages such as B cells (*NKBIA*[56]). Overall, the k-means edge clustering approach helped identify the key regulators with known or plausible roles in hematopoiesis that could explain the differences among the different lineages.

Our LDA topic analysis predicted several cell type-specific network components with different extents of conservation across the lineage (Fig. 6c, d, Supplementary Figs. 18–20). These topics were enriched in diverse biological processes such as cell cycle (Topic 1 and 8, Supplementary Fig. 21) and blood related processes (Topic 9). Topic 2 showed a gradual rewiring of an *ID2*-specific network from the HSC populations (C8, C3, C2), to *KLF1* and *MYC* centered networks for C7 and C10 which represented the CMP populations (Fig. 6c, d). *ID2* which belongs to the Inhibitors of DNA family of proteins has been shown to regulate both the erythroid and lymphoid lineages[57] and is consistent with its presence in the C8, C3, C2 clusters. Furthermore, *KLF1* connectivity was more pronounced in C7 compared to C10, which could indicate these cells are more committed than those in C10. Similarly, *PBX1* which is a key regulator of differentiation versus self-renewal was seen in C7 and C9. Topic 3 captured additional differences between the two GMP clusters, C6 and C9, with *IRF8* exhibiting more connections in C6 compared to C9 (Fig. 6d, Supplementary Fig. 18). Topics 1, 6 and 10 exhibited a conserved core around *HMGB2, TSC22D3*, and *YBX1* respectively, across all cells clusters (Supplementary Figs. 18–20). *HMGB2* is an important regulator for HSCs[58]. Both *YBX1* and *TSC22D3*, which were also identified in our k-means analysis, have known roles in hematopoiesis[48]. Topic 8 was associated with various cell cycle and chromatin remodeling regulators such as *TOP2A, CDC20,* and *CCNB1* (Supplementary Figs. 20, 21). Taken together, the LDA analysis identified subnetworks centered on candidate key regulators with known general roles in hematopoiesis as well as regulators involved in specific lineage decisions.

### Inferring gene regulatory networks in human fetal hematopoiesis

Our applications of scMTNI so far were on cell lineages where a branching structure was computationally inferred. To examine the utility of scMTNI in a system with known branching lineage structure, we applied it to a published scATAC-seq and scRNA-seq dataset of human fetal hematopoiesis[59], which captured specification to multiple blood lineages (Fig. 7a). We considered the cell populations measured with both scATAC-seq and scRNA-seq datasets at two resolutions: (1) coarse resolution comprising hematopoietic stem cell (HSC), multipotent progenitors (MPPs), lymphoid-myeloid progenitors (LMPs), MK-erythroid-mast progenitors (MEMPs), granulocytic progenitors

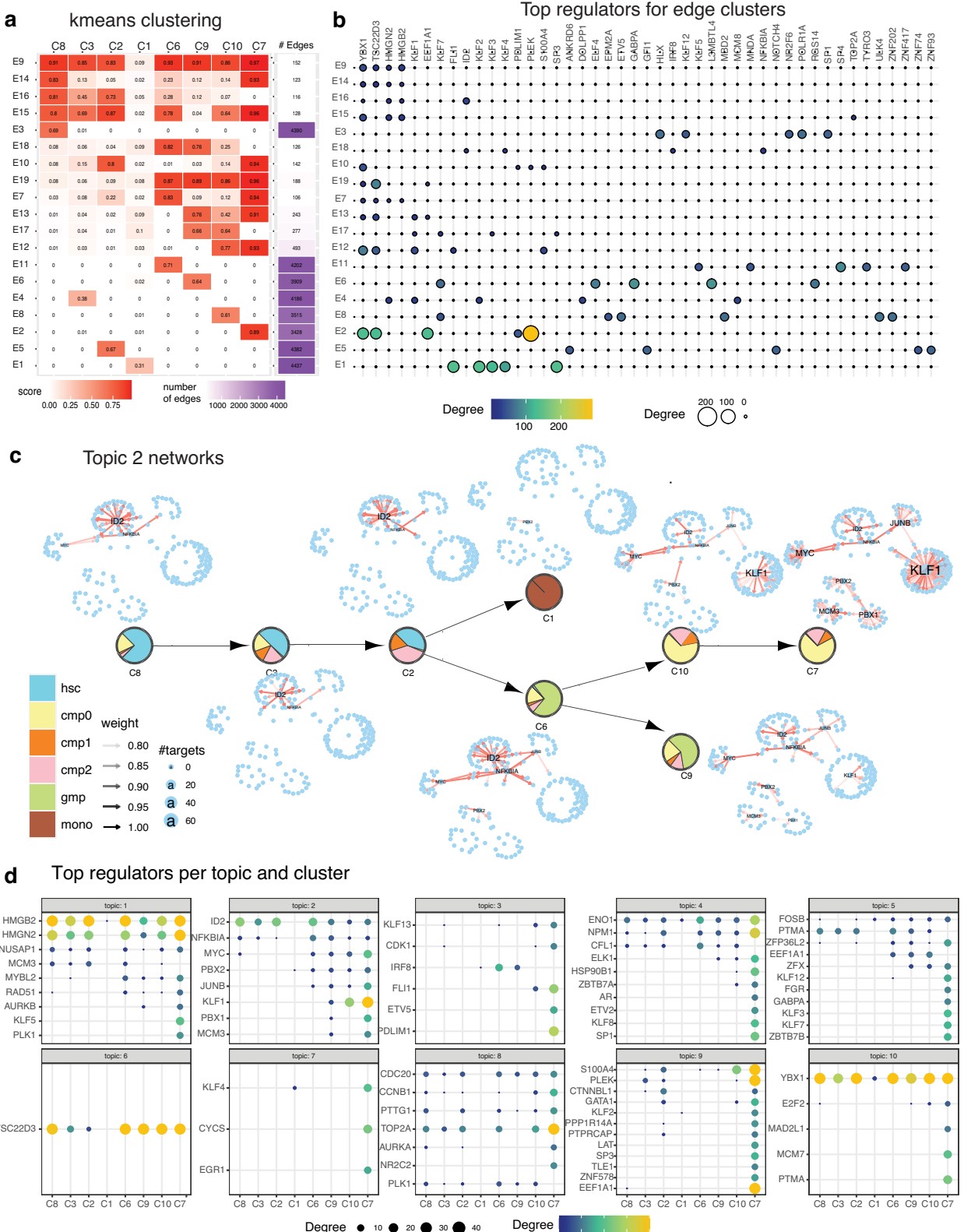

**Fig. 6 | Network rewiring during hematopoietic differentiation. a** k-means edge clusters of the top 5k edges (rows) across 8 cell clusters (columns). The edge confidence matrix was clustered into 19 clusters to identify common and divergent networks. The red intensity corresponds to the average confidence of edges in that cluster. Shown also are the number of edges in the edge cluster. **b** Top 5 regulators of each edge cluster. Shown are only regulators with at least 10 targets in a given edge cluster. The size and brightness (yellow) of the circle is proportional to the number of targets. **c** Topic-specific networks across each cell cluster for topic 2. The layout of each network is the same, edges present in a particular cell cluster are shown in red. Labeled nodes correspond to regulators with degree larger than 10. **d** Top regulators associated with each cell cluster's network in each topic. Shown are only regulators that have at least 10 targets in any cell cluster. The more yellow and larger the circle, the greater are the number of targets for the regulator. Source data are provided as a Source Data file.

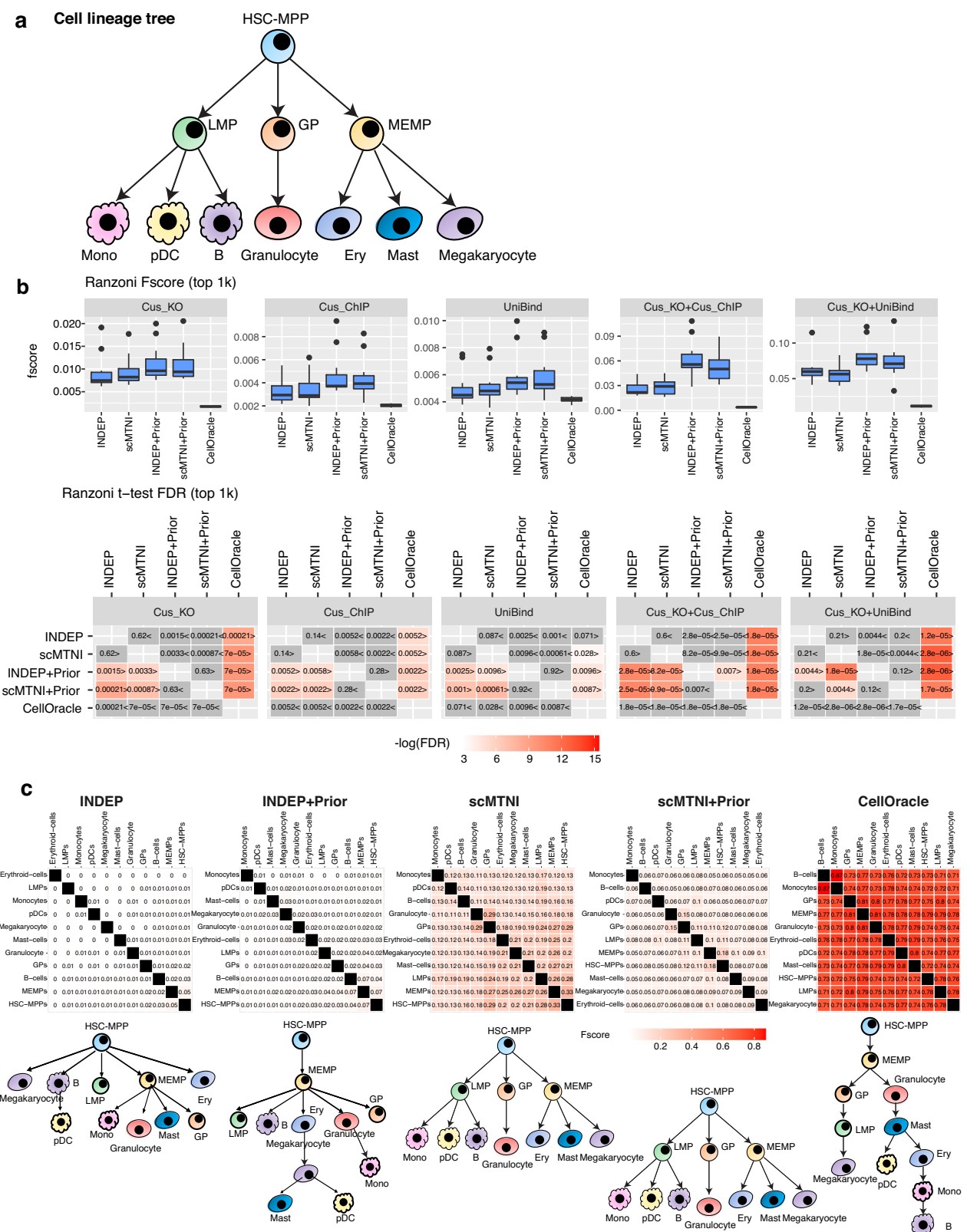

(GPs), and (2) fine-grained resolution, which additionally included the derived cell types from these progenitor populations. We evaluated the methods that incorporate prior and their no-prior versions on this dataset: scMTNI, scMTNI+Prior, INDEP, INDEP+Prior, and CellOracle, at two levels of resolution of the cell types (Methods).

On the fine lineage, algorithms that did not use priors (INDEP and scMTNI) performed comparably based on F-score (with no significant

difference) on all five gold standards (Fig. 7b, Supplementary Figs. 22, 23)). INDEP+Prior, scMTNI+Prior, which use priors were significantly better than methods without priors, while CellOracle performed the worst in all gold standards. INDEP+Prior and scMTNI+Prior are comparable across the gold standard datasets. Based on predictable TFs, scMTNI+Prior and INDEP+Prior were the best (Supplementary Fig. 24). As observed in the Buenrostro dataset, CellOracle did comparably to

**Fig. 7 | Inference of cell type-specific networks for human fetal hematopoiesis data. a** Cell lineage structure linking the cell clusters from scRNA-seq. **b** Boxplots showing F-scores of $n = 11$ cell clusters for top 1k edges in predicted networks from scMTNI, scMTNI+Prior, INDEP, INDEP+Prior, and CellOracle compared to gold standard datasets (top). FDR-corrected t-test to compare the F-score of the row algorithm to the F-score of the column algorithm (bottom). The two-sided paired t-test is conducted on F-scores of $n = 11$ cell clusters for every pair of algorithms. A FDR < 0.05 was considered significantly better. The sign < or > specifies whether the row algorithm's F-scores were worse or better than the column algorithm's F-scores. The color scale is specified for − log(FDR), with the red color proportional to significance. Non-significance is colored in gray. In the boxplot, the horizontal middle line of each plot is the median. The bounds of the box are 0.25 quantile ($Q_1$)

and 0.75 quantile ($Q_3$). The upper whisker is the minimum of the maximum value and $Q_3 + 1.5*IQR$, where $IQR = Q_3 − Q_1$. The lower whisker is the maximum of the minimum value and $Q_1 − 1.5*IQR$. **c.** Pairwise similarity of networks from each cell cluster using F-score on the top 5k edges. Rows and columns ordered by hierarchical clustering using F-score as the similarity measure. Reconstructed cell lineage trees are shown at the bottom of the pairwise F-score similarity matrix and are constructed using the MST algorithm on the F-score matrix. HSC-MPP hematopoietic stem cells and multipotent progenitors, LMP lymphoid-myeloid progenitors, MEMP MK-erythroid-mast progenitors combined with cycling MEMPs, GP granulocytic progenitors, Ery erythroid cells, Mono monocyte, pDC plasmacytoid dendritic cells. Source data are provided as a Source Data file.

other methods on the ChIP-based gold standards (Unibind, Cus_ChIP), but had fewer predictable TFs in the other gold standards. The poor performance of CellOracle is likely due to its complete reliance on the prior network for determining the structure of the final inferred network. We compared scMTNI+Prior and CellOracle on the coarse lineage and observed similar superior performance of scMTNI+Prior on both F-score and predictable TF metrics (Supplementary Fig. 30A, B).

We next examined the lineage structure by constructing an MST from pairwise distances of the inferred networks and compared it to the ground truth (Fig. 7c). The single-task learning methods INDEP and INDEP+Prior inferred networks had very low overlap for each pair of cell lines and the resulting lineage tree was different from the ground truth (Fig. 7c). In contrast, scMTNI and scMTNI+Prior were able to recover the cell lineage exactly as the input cell lineage tree. CellOracle, inferred more similarity across cell types and captured several aspects of the original lineage (e.g., MEMP deriving from HSC-MPP), but did not correctly recover several other aspects (e.g., LMPs and GPs derived from HSC, Granulocytes derived from GPs). For the coarse lineage, scMTNI+Prior and CellOracle inferred the same tree, but placed LMPs and GPs under MEMPs instead of under HSCs (Supplementary Fig. 30C). Taken together, these results show that scMTNI+Prior's framework of using lineage information and accessibility results in inference of more accurate GRN structure and dynamics during the differentiation process for known branching cell type trajectories.

## Examining dynamics of GRN components for fetal hematopoiesis

We applied our k-means and LDA analysis to identify regulators associated with edge rewiring and subnetwork changes for the fine (Fig. 8a–c, Supplementary Figs. 25–28) and coarse hematopoiesis lineages (Fig. 8d, Supplementary Figs. 31–35). The k-means analysis identified edge clusters spanning multiple cell types of the lineage tree (e.g., E16, E15, E21, E14, E13, E19, E7) as well as individual lineages (E4: B cells, E3: Granulocytes, E5: Erythrocytes, E9: Mast cells, E2: HSC-MPPs, E18: MEMPs) (Fig. 8a). We examined the regulators associated with the edge clusters shared across multiple cell types and found *HNRNPK* and *PTMA* to be frequently associated with these clusters (Fig. 8b). *HNRNPK* has a number of regulatory functions across diverse cell types including as a regulator of hematopoiesis[60]. *PTMA*, which stands for prothymosin alpha is not well understood for its function but is implicated in growth and survival of cells of hematopoietic origin, and required for the filament-inducing activity of macrophage lysate[61], which would be consistent with its expression in the hematopoietic lineage[62]. E17 had edges common to the Myeloid lineage spanning HSC-MPPs, MEMPs, Mast-cells, Megakaryocytes and Erythroid populations and had *ENO1, NPM1, SNRPD1* in addition to *HNRNPK* and *PTMA* as top regulators (Fig. 8b). *ENO1* encodes a glycolytic enzyme which is expressed in several human tissues and has been shown to be a regulatory enzyme with links to the MYC pathway[63]. E2 had edges specific to HSC-MPPs and was associated with

*PTMA, SNRPD1, SOX4* and *EEF1A1*, which have immune-related functions. E18 which was specific to MEMPs was associated with *KLF1, BRPF3* and *PTMA*. *KLF1*, which was found in the Buenrostro et al. dataset of adult hematopoiesis as well[45], is an essential regulator for the erythroid lineage[52,53], and was also found to be upregulated by Ranzoni et al. as cells transitioned from HSC/MPP to MEMPs[59]. E16 and E14 are edge clusters shared across all cell types with *EEF1A1, CDC20, HMGN2, NPM1, TOP2A* as top regulators. *HMGN2* belongs to the high-mobility group of proteins, which was identified in our analysis of the Buenrostro et al. dataset as well. Other regulators implicated cell cycle (*CDC20, TOP2A*) or more general regulators of development and proliferation (*NPM1*). Cell-cycle and cell-fate decisions are inherently tied especially in progenitor populations where the cell fate decision could be influenced by the cell cycle stage of the cells[64]. The k-means analysis of the coarse lineage exhibited much more shared network structure compared to the fine lineage, though it also identified edge sets specific to each coarse cell type (E1: HSC, E3: GPs, E2: LMPs, Supplementary Fig. 31). Several of the regulators identified in the fine lineage analysis were seen in the coarse lineage analysis showing overall consistency of our results. For example, E8 which had edges shared across all cell types had *EEF1A1, FOS, HMGN2, NPM1* as the top regulators. Similarly, *KLF1* was identified in the MEMP-specific edge cluster in the coarse (E4) and fine lineages (E17). The coarse lineage analysis also found additional regulators. For example, E2, which was specific to the LMP lineage was associated with *IRF8, KLF3, BAG4*, and *MAP2K7*. *IRF8*, which was identified in the Buenrostro et al. dataset as well plays a key role in innate immune response and is an essential for development of the lymphoid lineage including B cells[55], monocytes and pDCs[65].

Our LDA analysis identified topics representing subnetworks that rewire from the HSC state to different lineages (Methods). The topic genes were enriched in immune response (topic 1), cell-cycle (topics 2, 3 and 5), cellular respiration (topic 4) and general metabolic processes (topic 7, Supplementary Fig. 29A). LDA topic 3 identified a regulatory subnetwork that gained connections in B cells for regulators like *FOXP4* and *PPR2R5B* (Fig. 8c, Supplementary Fig. 26) and was enriched for cell cycle processes (Supplementary Fig. 29A). In contrast, topic 1 represented an opposite pattern of gradual loss of edges connected to *FOS* from HSC-MPP to downstream lineages (Supplementary Fig. 25). *FOS* was found to be upregulated in Ranzoni et al. in the HSCs/MPPs population[59]. Other topics exhibited conserved hubs like *PTMA* (topic 4, Supplementary Fig. 26), *HNRNPK* (topic 8, Supplementary Fig. 27)), and *NPM1* (topic 5, Supplementary Fig. 26) across multiple lineages and several cell cycle regulators such as *TOP2A* and *CDC20* (topic 2, Fig. 8c, Supplementary Fig. 25). On the coarse lineage, the LDA analysis revealed more hubs in HSC-MPPs which were lost when differentiating to the other lineages (Fig. 8d, Supplementary Figs. 31–35). The exceptions were *ENO1* (topic 7, Supplementary Fig. 34), *HMGN2* and *NPM1* (topic 4, Supplementary Figs. 31, 33) and *PTMA* (topic 3, Supplementary Fig. 31), which persisted at all lineages. *NPM1*, which was found both in fine and coarse tree, plays an important role in hematopoietic progenitors, especially in early myeloid differentiation[66]. A few regulators also gained connections in specific lineages, for

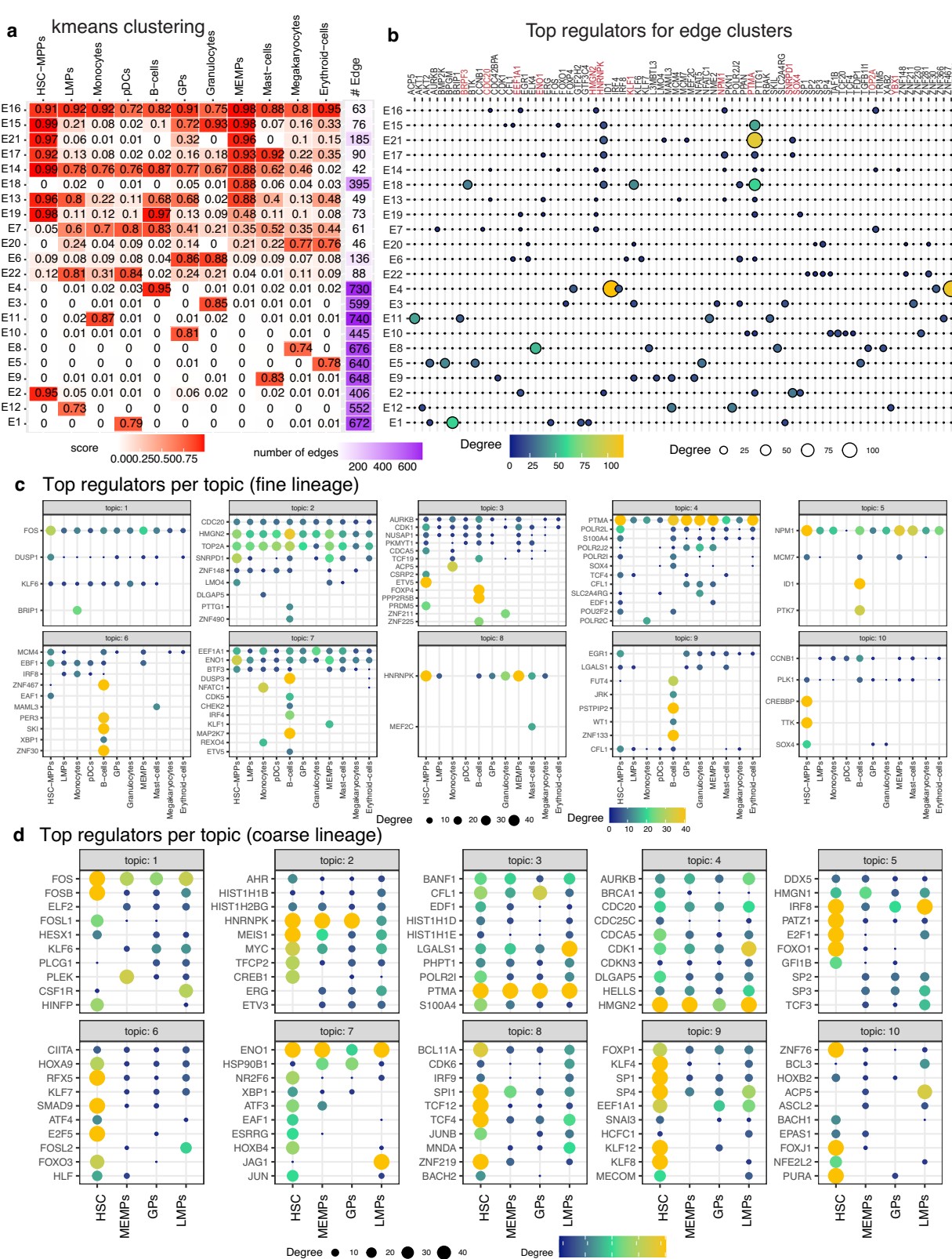

example, *LGALS1* (topic 3), *JAG1* (topic 7), *CDK1* (topic 4) had more edges in the LMP lineage and *PLEK* in the MEMP lineage (Supplementary Fig. 31). Both *LGALS1*[67] and *JAG1*[68] have been shown to be involved in hematopoiesis, however, the specific roles in this process is not as well-characterized. In topic 5, we observed the persistence of an *IRF8*-specific network from the HSCs/MPPs to LMPs populations, which was lost in MEMPs/GPs lineage and is consistent with our k-means analysis

and our results from Buenrostro et al. (Supplementary Fig. 33). Taken together, the k-means and LDA analysis identified several components of fetal hematopoiesis GRNs that changed as cells differentiated from HSC-MPP to differentiated cell types. While many of the regulators have well-characterized roles in hematopoiesis, several are previously uncharacterized that can be followed up with targeted functional studies.

**Fig. 8 | Network rewiring during human fetal hematopoiesis. a** k-means edge clusters of the top 1k edges (rows) across 11 cell clusters (columns). The edge confidence matrix was clustered into 21 clusters to identify common and divergent networks. The red intensity corresponds to the average confidence of edges in that cluster. Shown also are the number of edges in the edge cluster. **b** Top 5 regulators of each edge cluster. The size and brightness of the circle is proportional to the number of targets. Regulators mentioned in text are in red. **c** Top regulators associated with each cell cluster's network in each topic for fine-grained lineage tree. Shown are only regulators that have at least 10 targets in any cell cluster. The brighter and larger the circle, the greater are the number of targets for the regulator. **d** Top regulators associated with each cell cluster's network in each topic for coarse lineage tree. Shown are only regulators that have at least 10 targets in any cell cluster. The brighter and larger the circle, the greater are the number of targets for the regulator. For ease of interpretation only the top 10 regulators per topic are shown. The full list of regulators per topic are shown in Supplementary Fig. 31. Source data are provided as a Source Data file.

## Discussion

Single-cell technologies have transformed our ability to study cellular heterogeneity and cell-type specific gene regulation of known and novel cell populations. Defining gene regulatory networks from scRNA-seq data of developmental systems has remained challenging as most existing methods have assumed a static view of the GRN and do not leverage accessibility to inform the GRN structure. To address this need, we developed single-cell Multi-Task Network Inference (scMTNI), a probabilistic graphical model-based approach that uses multi-task learning to infer cell type-specific GRNs on a cell lineage tree by integrating scRNA-seq and scATAC-seq data and model the dynamics of these regulatory interactions on a lineage. A major benefit of the scMTNI framework is its flexibility in incorporating different sources of accessibility information as well as the ability to model dynamics on cell lineages of different topologies. The probabilistic prior-based framework makes scMTNI more robust to noisy or incomplete accessibility data and allows the incorporation of additional regulators such as signaling proteins and TFs with no binding information. Guided by the cell lineage structure, scMTNI's inferred networks exhibit meaningful changes along the trajectory and identify regulators and network components specific to cell populations transitioning to different lineage paths.

Multi-task learning is well-suited for the inference of cell type-specific GRNs. However, a key question is how to implement multi-task learning for GRN inference. A number of multi-task learning algorithms were developed for inferring GRNs and functional networks from bulk transcriptomic data but have not been systematically compared for their effectiveness on single-cell transcriptomic data. Some approaches, such as AMuSR[28] have used a flat hierarchy where all the tasks are considered equally related. For heterogeneously related datasets, a hierarchy or a tree is well-suited to model the dependence across datasets. Such hierarchies can be implemented as a phylogenetic tree with observed data at the tips of the tree as in GNAT[26] and MRTLE[25], or as a cell-lineage tree with observations at all nodes in the tree. scMTNI and MRTLE both use a tree-based structure prior, whereas AMuSR, GNAT, and Ontogenet used a regularized regression parameter to implement multi-task learning. scMTNI and MRTLE have better performance in predicting the gene regulatory relationships than single-task learning algorithms. The performance of Ontogenet is better than the single-task learning algorithms LASSO and INDEP in at least two cell types, and comparable to SCENIC. A prominent factor contributing to the difference in the performance of the algorithms was whether the models inferred a directed graph versus an undirected graph, with GNAT generally suffering likely due to this reason. Performance of GNAT is worst among multi-task learning algorithms and comparable to the single-task learning algorithms. We speculate that the undirected graphical models learned by GNAT might be a reason that the performance is not as good as other multi-task learning algorithms. We also examined the performance of algorithms across different parameter settings that control for sparsity as well as for sharing information. We found that the algorithms were generally robust to the setting of sharing and more sensitive to the extent of sparsity. However, multi-task learning algorithms generally outperformed single-task learning algorithms indicating that this is a useful direction for

methodological development for GRN inference from single cell omic datasets. Importantly, single-task learning infers very different networks that makes it challenging to study transitions across the networks.

Once GRNs are inferred across multiple cell types, the next challenge is to examine which components of the GRNs change along the lineage. We developed two complementary techniques to study dynamics. Our k-means edge clustering method was able to find regulatory connections that were unique to each cell cluster, while our LDA topic model-based dynamic network analysis highlighted sub-networks that were activated or deactivated along the lineage. We applied our tools to study GRN dynamics in adult and fetal hematopoietic cell differentiation and reprogramming from mouse embryonic fibroblasts to embryonic stem cells. We found that these systems exhibited different dynamics, with the reprogramming system exhibiting more edges shared across populations compared to the adult hematopoietic system which identified most edges as cell cluster-specific. In all three systems, our analysis identified known and previously uncharacterized regulators. For example, in the reprogramming system, we found that cells that were closer to the end point pluripotent state already had an *Esrrb*-centered GRN component active. In contrast, cells that were on an alternate trajectory exhibited persistence of the MEF regulatory program including regulators such as *Aebp1*. Between adult and fetal hematopoiesis we found several shared regulators that were known lineage-specific regulators (e.g., *IRF8* in the lymphoid lineage), but also identified regulators unique to each system which could be followed up with future validation studies.

scMTNI currently assumes that the input lineage structure is accurate. However, lineage construction, especially from integrated scRNA-seq and scATAC-seq datasets is a challenging problem. One direction of future work is to assume the initial lineage structure is inaccurate and incorporate the refinement of the lineage structure as part of the GRN inference procedure. A second direction of work is to model more fine-grained transitions within each cell population, for example using RNA velocity or pseudotime[69], which will complement the coarse-grained dynamics that scMTNI currently handles. Studies from bulk RNA-seq data have shown that estimating hidden transcription factor activity (TFA)[70] can further improve the performance of network inference. Thus, another direction of future work is to estimate hidden TFA and incorporate these to improve the accuracy of the inferred networks. Finally, SCENIC generally outperforms the single-task learning algorithms which do not use prior, which is likely because of its regression-tree based model that captures non-linear dependencies and is less prone to the sparsity of the dataset. While scMTNI's stability selection framework can capture some non-linearities, another direction of future work is to extend scMTNI to model more non-linear dependencies.

In summary, scMTNI is a tool to infer cell type-specific regulatory networks and their dynamics on a cell lineage which combines scRNA-seq and scATAC-seq data. As single cell multi-omic datasets become increasingly available, we expect scMTNI to be broadly applicable to predict GRNs and prioritize regulators associated with regulatory network dynamics across cell types in diverse cell-fate specification processes.

## Methods

This research complies with all relevant ethical regulations. Mice used in the reprogramming study were maintained in agreement with our UW-Madison Institutional Animal Care and Use Committee (IACUC) approved protocol (ID M005180-R03).

### Single-cell Multi-Task Network Inference (scMTNI)

Single-cell Multi-Task Network Inference (scMTNI) is a probabilistic graphical model-based approach that uses multi-task learning to infer gene regulatory networks for cell types related by a cell lineage tree (Fig. 1). We define a cell type to be a group of cells with similar transcriptome and accessibility levels as defined by existing cell clustering methods. Each task learns the gene regulatory network (GRN), $\mathbf{G}^{(d)}$ for each cell type or cell cluster $d$. Given cell type-specific datasets for $M$ cell types, $\mathbf{D} = \{\mathbf{D}^{(1)}, \cdots, \mathbf{D}^{(M)}\}$, our task is to find the set of graphs $\mathbf{G} = \{\mathbf{G}^{(1)}, \cdots, \mathbf{G}^{(M)}\}$ and parameters $\mathbf{\Theta} = \{\theta^{(1)}, \cdots, \theta^{(M)}\}$ for each of the cell types. $\mathbf{G}^{(d)}$ is modeled as a dependency network[22], a class of probabilistic graphical models for inferring directed, predictive relationships among random variables (regulators and genes). Each gene is modeled as a random variable $X_i^{(d)}$ which encodes the expression level of gene $i$ in each cell. A conditional probability distribution $P(X_i^{(d)}|\mathbf{R}_i^{(d)})$ models the relationship between gene $i$ and its set of regulators, $\mathbf{R}_i^{(d)}$ in cell type $d$. In a dependency network, GRN inference entails estimating the regulators $\mathbf{R}_i^{(d)}$ for each gene $i$ in each cell type $d$. To enable joint learning of these cell type-specific networks, our goal is to find the set $\mathbf{G} = \{\mathbf{G}^{(1)}, \cdots, \mathbf{G}^{(M)}\}$ and parameters $\mathbf{\Theta} = \{\theta^{(1)}, \cdots, \theta^{(M)}\}$ by estimating the posterior distribution of these two sets and finding their maximum a posteriori values:

$$P(\mathbf{G},\mathbf{\Theta}|\mathbf{D}) \propto P(\mathbf{D}|\mathbf{G},\mathbf{\Theta})P(\mathbf{\Theta}|\mathbf{G})P(\mathbf{G}) \tag{1}$$

$P(\mathbf{D}|\mathbf{G},\mathbf{\Theta})$ is the data likelihood, expanded as $\prod_d P(\mathbf{D}^{(d)}|\mathbf{G}^{(d)}, \theta^{(d)})$. In a dependency network, pseudo likelihood[22] is used to approximate the data likelihood for each cell type, defined as the products of the conditional distribution of each random variable $X_i^{(d)}$ given its neighbor set $\mathbf{R}_i^{(d)}$ in cell type $d$, $P(X_i^{(d)}|\mathbf{R}_i^{(d)}, \theta_i^{(d)})$. Thus, the likelihood can be written as:

$$P(\mathbf{D}|\mathbf{G},\mathbf{\Theta}) \propto \prod_{d\in\{1,\dots,M\}} \prod_{i\in\{1,\dots,N\}} P(X_i^{(d)}|\mathbf{R}_i^{(d)},\theta_i^{(d)}) \tag{2}$$

Given the neighbor set $\mathbf{R}_i^{(d)}$, the above quantity can be computed efficiently. We assume that each variable $X_i^{(d)}$ and its neighbor set $\mathbf{R}_i^{(d)}$ in cell type $d$ are from a multi-variate Gaussian distribution. Thus, $P(X_i^{(d)}|\mathbf{R}_i^{(d)},\theta_i^{(d)})$ can be modeled using a conditional Gaussian distribution with mean $\mu_{X_i^d|R_i^d}$ and variance $\sigma^2_{X_i^d|R_i^d}$ which can be estimated in closed form. $\mathbf{R}_i^{(d)}$ is selected from the input list of regulators using a greedy search algorithm, executed in parallel across all cell types (See Supplementary Methods). The second term $P(\mathbf{\Theta}|\mathbf{G})$ in Equation (1) is estimated using the maximum likelihood settings of the parameters. The third term $P(\mathbf{G}) = P(\mathbf{G}^{(1)}, \cdots, \mathbf{G}^{(M)})$ in the objective function is the structure prior and is defined in a way to capture the state of an edge across all cell types modeled, where $\mathbf{G} = \{\mathbf{G}^{(1)}, \cdots, \mathbf{G}^{(M)}\}$. We assume that $P(\mathbf{G})$ is composed of two priors, one is the cell-type specific prior $P(\mathbf{T})$, where $\mathbf{T} = \{T^{(1)}, \dots, T^{(M)}\}$, and the other one is a cell lineage structure prior $P(\mathbf{S})$ which captures the similarity between related cell types along the cell lineage tree, where $\mathbf{S} = \{S^{(1)}, \dots, S^{(M)}\}$.

$P(\mathbf{T})$ is the cell-type specific prior, which decomposes over a product of cell-type specific graphs: $P(T^{(1)}, \dots, T^{(M)}) = \prod_{d=1}^{M} P(T^{(d)})$. The $P(T^{(d)})$ decomposes over a product of individual edge configurations, $P(I_{u,v}^{(d)})$, where $I_{u,v}^{(d)}$ is an indicator function that represents whether there exists an edge between regulator $u$ to target gene $v$ in cell type $d$,

$X_u \to X_v$ as follows:

$$I_{u,v}^{(d)} = \begin{cases} 1, & \text{if there is an edge from } u \text{ to } v \text{ in cell type } d, \\ 0, & \text{otherwise}. \end{cases} \tag{3}$$

As in Roy et al.[71], we model the prior probability using a logistic function:

$$P\left(I_{u,v}^{(d)} = 1\right) = \frac{1}{1 + e^{-(\beta_0 + \beta_1 * m_{uv}^{(d)})}} \tag{4}$$

The $\beta_0$ parameter is a sparsity prior that controls the penalty of adding of a new edge to the network, which takes a negative value ($\beta_0 < 0$). A smaller value of $\beta_0$ will result in a higher penalty on adding new edges and will therefore infer sparser networks. The $\beta_1$ parameter controls how strongly motifs are incorporated as prior ($\beta_1 \geq 0$). A higher value of $\beta_1$ will result in motif presence being valued more strongly to select an edge. $\beta_1$ is set to 0 when there is no cell type-specific motif information available. $m_{uv}^{(d)}$ is the weight of the edge from regulator $u$ to target $v$ in the prior network and is computed based on the motif instance score if gene $v$ has a motif instance of regulator $u$ in its promoter region, additionally filtered by available bulk or single cell ATAC-seq peaks. Thus, we have

$$P(\mathbf{T}) = \prod_{d=1}^{M} P(T^{(d)}) = \prod_{d=1}^{M} \prod_{u,v;u\neq v} P(I_{u,v}^{(d)}) \tag{5}$$

The cell lineage structure prior $P(\mathbf{S})$ is constructed to make use of multi-task learning. We define $P(S^{(1)}, \dots, S^{(M)})$ as a product over a set of edges between regulators and target genes: $\prod_{u,v;u\neq v} P(I_{u,v}^{(1)}, \dots, I_{u,v}^{(M)})$. Under the assumption that the prior probability of the edge state in one cell type is only dependent upon its state in the predecessor cell type, we have:

$$P(\mathbf{S}) = \prod_{u,v;u\neq v} P(I_{u,v}^{(1)}, \dots, I_{u,v}^{(M)}) = \prod_{u,v;u\neq v} \prod_{d\in\{1,\dots,M\}} P(I_{u,v}^{(d)}|I_{u,v}^{pa(d)}) P(I_{u,v}^{(r)}), \tag{6}$$

where $pa(d)$ denotes the predecessor cell type of cell type $d$ on the cell lineage tree and $r$ denotes the starting root cell type. $P(I_{u,v}^{(d)}|I_{u,v}^{pa(d)})$ is a measure of overall gain and loss of regulatory connections between related cell types and is assumed to be the same across the set of edges. Thus, it can specified by three parameters: the probability of gaining a regulatory edge in the root cell type, $p_r = P(I_{u,v}^{(r)})$, the probability of gaining a regulatory edge in cell type $d$ given that the edge does not exist in its predecessor cell type, $p_g^{(d)} = P(I_{u,v}^{(d)} = 1|I_{u,v}^{pa(d)} = 0)$, and the probability of maintaining a regulatory edge in cell type $d$, given it is present in its predecessor cell type $p_m^{(d)} = P(I_{u,v}^{(d)} = 1|I_{u,v}^{pa(d)} = 1)$. These parameters of the priors can be set by the user or estimated empirically by analyzing different configurations and selecting those values with the best agreement with existing biological knowledge of the system. scMTNI uses a greedy score-based structure learning algorithm. Please refer to Supplementary Methods for details.

### Input datasets

**Simulated datasets.** To benchmark the performance of different multi-task and single-task learning algorithms, we simulated single cell expression data from a lineage resembling a linear differentiation process for three cell types (Fig. 2a). We simulated network dynamics on the lineage while controlling the extent of similarity with the three prior parameters: $p_r$, the probability of having an edge in the starting/root cell type; $p_g^{(d)}$, the probability of gaining an edge in cell type $d$ that is not in the predecessor cell type; $p_m^{(d)}$, the probability of maintaining an edge in cell type $d$ from the predecessor cell type. We set $p_r = 0.5, p_g^{(d)} = 0.4$ and $p_m^{(d)} = 0.7$ or $0.8$ and simulated three networks from a linear lineage tree for each of the three cell types, each with 15

regulators and 65 genes. Next, we applied BoolODE on the simulated gene regulatory networks and generated single cell expression data for 2000 cells for each cell type. To mimic the dropouts in the scRNA-seq data, we added 80% sparsity uniformly to all genes on the simulation data. We refer to this simulated dataset as dataset 1, consisting of 65 genes and 2000 cells for three cell types. We generated smaller sample sizes of these datasets, dataset 2 and dataset 3 by downsampling dataset 1 to 1000 cells (dataset 2) and 200 cells (dataset 3). We applied each of the algorithms on these three datasets within a stability selection framework and evaluated their performance based on AUPR and F-score as described in the Evaluation section.

**Human hematopoietic differentiation data.** Buenrostro et al.[45] measured single-cell accessibility (scATAC-seq) and single-cell RNA sequencing (scRNA-seq) data to study the regulatory dynamics during human hematopoietic differentiation for multiple immunophenotypic cell types: hematopoietic stem cells (HSCs), common myeloid progenitors (CMPs) and granulocyte-macrophage progenitors (GMPs) and Monocytes (Mono). We downloaded processed scRNA-seq data for each cell type from Data S2 of Buenrostro et al. (https://ars.els-cdn.com/content/image/1-s2.0-S009286741830446X-mmc4.zip) and fragment files for the scATAC-seq data from Chen et al.[72] (https://github.com/pinellolab/scATAC-benchmarking/tree/master/Real_Data/Buenrostro_2018). For the scATAC-seq data, we mapped the fragments into 23,347,540 bins with length of 1000bp. Next, we mapped 1 kb bins to the nearest gene and extracted cells with cell barcodes labeled as HSC, CMP, GMP, and Mono. Next, we filtered out genes with sum of counts in all samples less than 100, producing a processed scATAC-seq dataset with 54,344 genes and 1315 cells across the four cell types. We extracted the count matrix of scRNA-seq from these four cell types; note that CMP cells were in three different clusters: CMP0, CMP1, and CMP2. After filtering out genes with non-zero expression in less than 5 cells, the scRNA-seq data had 12,558 genes and 4165 cells. We normalized the count matrix for depth and variance stabilization based on the pagoda pipeline[73]. We kept 12,393 common genes between scATAC-seq and scRNA-seq data and applied LIGER[23] to define integrated cell populations. We applied LIGER with $k \in 8, 10, 12, 15, 20$ factors and found $k = 10$ to be most appropriate. Cluster C8 was mainly composed of HSCs, C6 was mainly composed of GMP cells, C7 was mainly CMP0 cells, C1 was composed of Monocyte cells, and the rest of the clusters were a combination of several cell types. C5 had too few RNA cells (22 cells) so we excluded it from further analysis. Since the composition of C1 (73 cells) and C4 (37 cells) are very similar, mainly GMP and Mono cells, we combined these two clusters as C1. We inferred a cell lineage tree from the 8 cell clusters using a minimal spanning tree (MST) approach using the python package `scipy.-sparse.csgraph`. Briefly, we used the mean expression profiles across samples of these cell clusters and computed the Euclidean distance between every pair of cell clusters. Then, we inferred the MST from the distance matrix using `scipy.sparse.csgraph`.

To derive the prior network for each cell cluster we created cluster-specific bam files from the scATAC-seq data using the LIGER clusters. We pooled these bam files to generate pseudo bulk accessibility coverage and applied MACS2 (v2.1.0) to identify scATAC-seq peaks for each cell cluster[74]. We obtained sequence-specific motifs from the Cis-BP database (http://cisbp.ccbr.utoronto.ca/)[75] and used the script `pwmmatch.exact.r` available from the PIQ toolkit[76] to identify significant motif instances genome-wide using the human genome assembly of hg19. We mapped motifs to each scATAC-seq peak and mapped the peak to a gene if it was within ± 5000 bp of the transcription start site (TSS) of a gene. In this case, we connect all motifs to a TSS that are mapped to the same scATAC-seq peak. We used the maximum motif score from `pwmmatch.exact.r` for each motif-TSS pair and took the maximum value among all TSSs of a gene as the value for each motif-gene pair. The motif instance score is the

log ratio of the Position Weight Matrix (PWM) match score to a uniform background. Finally, to generate the edge weight for each TF-gene pair, we used the max score among all motifs mapped to the same TF. To normalize the edge weights across TFs, we converted these weights into percentile scores and selected the top 20% of edges as prior edges.

**Mouse cellular reprogramming data.** We generated an scATAC-seq time course dataset for cellular reprogramming from mouse embryonic fibroblast (MEFs) to induced pluripotent cells (iPSCs). The dataset contains a total of 6 time points corresponding to the starting MEF, the end pluripotent state (mESC), and four intermediate time-points of day 3, day 6, day 9 and day 12. The mice used to generate the MEFs used for reprogramming were housed in a facility that ran a 12 h light/12 h dark cycle, had an ambient temperature 72 °F and maintained humidity between 20–50%. Mice were maintained in agreement with our UW-Madison Institutional Animal Care and Use Committee (IACUC) approved protocol (ID M005180-R03). Male and female mice of breeding age (at least 6–8 weeks old) from a mixed 129/Bl6 background that are homozygous for the *Oct4-2A-Klf4-2A-IRES-Sox2-2A-c-Myc* (OKSM) transgene at the *Col1a1* locus and heterozygous for the reverse tetracycline transactivator (rtTA) allele at the *Rosa26* locus were time-mated, from which MEFs were isolated at E13.5. On E13.5, the pregnant female mouse is carefully dissected and all embryos are removed. The head and neck region of the embryo is separated from the rest of the body and any organ tissues present are also removed, leaving only the fibroblasts. The remaining fibroblast tissue is emulsified and plated onto a 15 cm. The cells are passaged 1–2 additional times before being collected and stored in liquid nitrogen until the start of the experiment. In this study, MEFs with a homozygous genotype for the OSKM transgene and rtTA allele were used for reprogramming experiments. Male neonatal human foreskin fibroblasts (HFFs) from American Type Culture Collection (HFF-1 SCRC-1041) were used as feeders for our reprogramming cells. HFFs were passaged and expanded ~5 times prior to being irradiated. HFFs were irradiated at a level of 80 Gray prior to being used as feeders for the reprogramming MEFs. The process of somatic cell reprogramming is unaffected and is not influenced by the sex of the starting cell population, so the sex of the MEFs used in this experiment is unknown as it is irrelevant to the observed results.

On Day -2, E13.5 reprogrammable MEFs were thawed and on Day -1, they were plated in gelatinized 6-well plates at a seeding density of 5000 cells per well. Reprogramming was induced on Day 0 by adding 2 ug/ml doxycycline (Sigma-Aldrich D9891) to each well, which induced OKSM expression, as well as irradiated DR4 feeder MEFs. Reprogramming cells were maintained in ESC media (knockout DMEM (Gibco #10829-018), 15% FBS (Biowest S1620), L-glutamine (Gibco #15140-122), Pen/Strep (Gibco #33050-061), NEAA (Gibco #11140-050), 2-mercaptoethanol (Sigma-Aldrich #M6250) and leukemia inhibitory factor (Sigma-Aldrich #L5158)). Media was changed every two days. Cells were collected and prepared in a single-cell suspension on days 3, 6, 9, and 12. To generate single-cell suspensions, cells in the wells were washed 5X with DPBS (Gibco #14190-144) and dissociated from plate using 0.25% Trypsin-EDTA (Gibco #25200-072). Trypsin was neutralized with soybean trypsin inhibitor (Sigma-Aldrich #T6522), cells were filtered through a 40um filter, and spun down for 3min at 300xg (RT). Cells were then resuspended in 1ml of 0.1% BSA-PBS (prepared by diluting 7.5% Bovine Albumin Fraction V solution (Gibco #15260-037) to 0.1% with DPBS) and pipetted up and down 50X. 6 ml of 0.1% BSA-PBS were added to cells and spun down again at $300 \times g$ for 3 min. Cells were finally resuspended in 1 ml of 0.1% BSA-PBS. Cell concentration was determined using an Invitrogen Countess II cell counter prior to nuclei isolation, transposition, and single-cell ATAC-sequencing.

scATAC-seq data were generated using the 10x Genomics platform with a targeted nuclei recovery of 4000 and targeted read depth

of 25k reads per nucleus. Sequencing was performed using the Illumina NovaSeq 6000 machine and samples were loaded onto a S1 flow cell. The scATAC-seq data was first processed through CellRanger ATAC pipeline (version 1.1.0) to provide the fragments file. We binned the genome at non-overlapping 1 kb bin and computed the number of fragments mapped to each 1 kb bin. Next, we mapped 1 kb bins to the nearest gene for all of the samples. The processed scATAC-seq data contains 25,824 genes and 30,344 cells.

We downloaded scRNA-seq datasets (GEO: GSE108222) for the same time points from ref. 32. We concatenated the expression data from two replicates at each time point and normalized the concatenated matrix for depth and variance stabilization based on a simplified implementation of the pagoda pipeline[73]. Next, for each time point, we removed genes with expression in less than 5 cells. We took the union of genes among all time points and concatenated the expression data across all time points as our final scRNA-seq data matrix. The processed scRNA-seq dataset contains 14,953 genes and 3460 cells. We had a total of 11,926 genes in common between the two datasets, which were used for downstream analysis. We applied LIGER with $k \in 8, 10, 12, 15, 20$ and found $k = 8$ to provide the optimal clustering of the scRNA-seq and scATAC-seq data determined based on the clustering of the accessibility and transcriptome of the MEF and ESC time points. We inferred a minimal spanning tree from the distance matrix of the pseudobulk expression profiles of each cluster using `scipy.sparse.csgraph`, similar to the Buenrostro et al. hematopoiesis dataset, and used it as the cell lineage tree. The prior motif was generated in the same way as for the hematopoiesis differentiation dataset using motifs for mouse from the CisBP database[75]. We used the 10 mm mouse genome assembly for this analysis.

**Human fetal hematopoietic differentiation data.** Ranzoni et al.[77] measured scRNA-seq and scATAC-seq data to study the regulatory dynamics during human developmental hematopoiesis for multiple immunophenotypic blood cell types from fetal liver and bone marrow. We obtained the scRNA-seq (gene by cell) and scATAC-seq data (peak by cell) matrices from https://gitlab.com/cvejic-group/integrative-scrna-scatac-human-foetal. We used the annotated cell clusters in ref. 77 for the scRNA-seq data: HSCs/MPPs combined with cycling HSCs/MPPs (HSCs-MPPs), lymphoid-myeloid progenitors (LMPs), MK-erythroid-mast progenitors combined with cycling MEMPs (MEMPs), granulocytic progenitors (GPs), granulocytes, erythroid cells, megakaryocytes, mast cells, monocytes, plasmacytoid dendritic cells (pDCs) and B cells. We took the union of genes among all cell types and concatenated the expression data as our final scRNA-seq data matrix. We normalized this concatenated matrix for depth and performed variance stabilization based on the pagoda pipeline[73] and removed genes with expression in less than 20 cells. The labeling provided by Ranzoni et al. for the scATAC-seq data omitted many of these cell types making it challenging to determine cell-type specific priors. To overcome this challenge we utilized a label transfer technique based on the method provided in the Seurat v3 package[78]. Briefly, we embedded the scRNA-seq and scATAC-seq cells (after mapping peaks to gene promoters) into a shared lower dimensional embedding ($k = 10$) utilizing LIGER[23]. We next defined "anchors", which are pairs of cells that provide a correspondence between the scRNA-seq and scATAC-seq modalities. Each anchor is defined as a mutual nearest neighbor in the lower dimensional space and has an anchor score computed based on the overlap of within and between dataset neighborhoods as specified in the Seurat v3 package. Once the anchor scores are established, we computed the anchor weights for each cell in the scATAC-seq data and transferred labels based on a linear combination of the anchor weights and labels associated with the scRNA-seq cells. Each scATAC-seq cell with a label score greater than 0.3 was assigned the maximally scoring label. Cells with score below 0.3 were not used to generate the prior network.

To derive the prior network for each cell type, we extracted scATAC-seq peaks present in each cell type derived from our label transfer method. For LMPs, as there are no cells in the scATAC-seq data labeled as LMPs, we took the union of peaks across LMP's derived cell types (monocytes, pDCs, and B cells) as the scATAC-seq peaks for LMPs. We used a similar strategy as the Buenrostro et al. dataset to generate the prior network. Briefly, we used the same sequence-specific motifs from the Cis-BP database[75] as the Buenrostro et al. data, mapped motifs to each scATAC-seq peak and mapped the peak to a gene if it was within ± 5000 bp of the gene TSS. For the coarse cell lineage tree, we merged all derived cell types from each parent cell type to produce four cell populations as follows: monocytes, pDCs, NK cells and B cells were merged with the LMP cells; erythroid cells, megakaryocytes, and mast cells were merged with MEMPs; and Granulocytes were merged with GPs. We applied the same approach as the fine tree to prepare the scRNA-seq expression data and prior networks for each cell type using union of scATAC-seq peaks in each cell type and its derived cell types.

## Application of network inference algorithms on simulated datasets

We used the simulated datasets to perform benchmarking of the different network inference algorithms. We also used this dataset to study the sensitivity of the algorithms to the different parameter settings. Below we describe each of the algorithms as well as the parameters used for each of the algorithms for the simulated datasets. For all three simulation datasets, we applied all algorithms other than SCENIC within a stability selection framework to estimate the confidence score for each edge in the predicted networks. For stability selection, we subsampled each dataset 20 times randomly using half of the cells and all genes. SCENIC has its own internal sub-sampling and directly outputs the edge importance. scMTNI and baseline methods require list of regulators and target genes information as input. This information is provided to all methods under comparison.

scMTNI: scMTNI has five hyper-parameters: $p_r$, probability of having an edge in the starting cell type; $p_g^{(d)}$, probability of gaining an edge in a child cell type $d$; $p_m^{(d)}$ the probability of maintaining an edge in $d$ from its immediate predecessor cell type; a sparsity penalty $\beta_0$, that controls penalty for adding edges; $\beta_1$, that controls the strength of incorporating prior network. We tested different configurations of the hyper-parameters: $p_r \in \{0.1, 0.15, 0.2, 0.25, 0.3, 0.35, 0.4, 0.45, 0.5\}$, and $p_g^{(d)} \in \{0.05, 0.1, 0.15, 0.2, 0.25, 0.3, 0.35, 0.4, 0.45\}$, and $p_m^{(d)} \in \{0.55, 0.6, 0.65, 0.7, 0.75, 0.8, 0.85, 0.9\}$, $\beta_0 \in \{-0.005, -0.01, -0.05, -0.1, -0.5\}$. $\beta_1$ was set to 0 as there is no prior network in the simulations. If the size of the predicted network for a parameter setting was smaller than the size of the simulated network, we disregarded this parameter setting for comparison. We used the area under the precision-recall curve (AUPR) to compare the scMTNI inferred networks to simulated networks. We also computed F-score on top $K$ edges ranked by the confidence score (where $K$ is the number of edges in the simulated network, C1: $K = 202$, C2: $K = 217$, C3: $K = 239$). Overall performance of scMTNI was stable across different parameter configurations (Supplementary Fig. 36, Supplementary Methods). To compare against methods, we used values from the best parameter settings for each dataset and cell type as well as all parameter settings (Supplementary Figs. 1, 2).

MRTLE: Multi-species regulatory network learning (MRTLE)[25] is a probabilistic graphical model-based algorithm that uses phylogenetic structure, transcriptomic data for multiple species, and sequence-specific motifs to infer the genome-scale regulatory networks across these species simultaneously. It was developed for bulk transcriptomic data and uses a dependency network model to specify the directed relationship among regulators to target genes. Sequence-specific motif instances can be incorporated as prior knowledge to favor edges supported with the presence of motifs. The multi-task learning

framework is embedded in the phylogenetic prior, which captures the evolutionary dynamics of regulatory edge gain and loss guided by the phylogenetic structure. The MRTLE algorithm has four parameters: $p_g$, the probability of gaining an edge in a child species $s$ that is not in the ancestor species; $p_m$, the probability of maintaining an edge in a species $s$ given it is also in $s$'s immediate ancestor of $s$; $\beta_0$, a sparsity penalty that controls penalty for adding edges, and a penalty $\beta_1$ that controls the strength of motif prior. In the simulation case, we examined different parameter configurations: $p_g \in \{0.05, 0.1, 0.15, 0.2, 0.3, 0.4\}$, $p_m \in \{0.5, 0.55, 0.6, 0.65, 0.7, 0.75, 0.8, 0.85\}$, $\beta_0 \in \{-0.005, -0.01, -0.05, -0.1, -0.5, -1\}$. $\beta_1$ was set to 0. The overall performance of MRTLE was stable across different parameter configurations (Supplementary Fig. 37). Similar to scMTNI, we used the AUPR and F-score of top $K$ edges to select the best parameter setting. The best parameter setting and all parameter settings were used to compare against other algorithms.

GNAT: The GNAT[26] algorithm uses a hierarchy of tissues to share information between related tissue and infers tissue-specific gene co-expression networks. It was developed for bulk transcriptomic data. GNAT models each network using a Gaussian Markov Random Field (GMRF). It has two parameters: the $L_1$ penalty $\lambda_s$ that controls the sparsity of the network, and the $L_2$ penalty $\lambda_p$ that encourages the precision matrix of children to be similar to its parent precision matrix. It initially learns a co-expression network for each leaf tissue. Then it infers the networks in internal nodes using the networks in the leaf nodes and updates the networks in leaf nodes iteratively until convergence. Since GNAT learns undirected networks, we transformed them to directed networks by adding edges from a regulator to a target. If the nodes of an edge are both candidate regulators, we output the edge in both directions. We tested different parameter configurations of $\lambda_s$ and $\lambda_p$. For data 1 ($n = 2000$), $\lambda_s$ were set to {30, 31, 32,..., 37}, and $\lambda_p$ were set to {30, 31, 32,..., 40}. For data 2 ($n = 1000$), $\lambda_s$ were set to {18, 19,..., 22}, and $\lambda_p$ were set to {18, 19,..., 25}. For data 3 ($n = 200$), $\lambda_s$ were set to {5, 6, 7, 8}, and $\lambda_p$ were set to {5, 6, 7, 8}. We found that $\lambda_s$ dominates the performance and under the same $\lambda_s$, changing $\lambda_p$ does not change the performance substantially (Supplementary Fig. 38). If the size of the predicted network for a parameter setting is smaller than the size of the simulated network, we removed this parameter setting. The ranges of $\lambda_s$ and $\lambda_p$ are slightly different and varying across different datasets. We used AUPR and F-score of top $K$ edges to select the best parameter settings. We compared the algorithms using the best and all parameter settings.

Ontogenet: The Ontogenet[27] algorithm was developed to reconstruct lineage-specific regulatory networks using cell type-specific gene expression data across cell lineages. It was developed for bulk transcriptomic data. To infer the regulatory networks for each cell type, Ontogenet uses a fused LASSO framework combined with an additional $L_2$ penalty. The $L_1$ penalty is introduced to control the sparsity of regulators, while the $L_2$ penalty is used to select correlated predictors. The multi-task learning uses a fused LASSO framework with an additional $L_1$ penalty on the difference of the regression weight of related cell types, which encourage the consistency of regulatory programs between related cell types. The Ontogenet algorithm has three parameters: the $L_1$ penalty $\lambda$ that controls the sparsity of the network, the $L_2$ penalty $\kappa$ that handles correlated predictors, and $\gamma$ that encourages the similarity of regulatory programs between related cell types. We tested different parameter configurations of $\lambda$, $\gamma$ and $\kappa$. For data 1 ($n = 2000$), $\lambda$ were set to {1000, 1250, 1500, 1750, 2000, 2250, 2500}, and $\gamma$ were set to {1000, 1250, 1500, 1750, 2000, 2250, 2500}. For data 2 ($n = 1000$), $\lambda$ were set to {500, 1000, 2000, 3000}, and $\gamma$ were set to {500, 1000, 2000, 3000}. For data 3 ($n = 200$), $\lambda$ were set to {475, 500, 525}, and $\gamma$ were set to {475, 500, 525}. $\kappa$ was set to {1, 5, 10} for each of the datasets. We found that $\lambda$ and $\gamma$ dominate the performance, while changing $\kappa$ does not change the performance significantly (Supplementary Fig. 39). If the size of the predicted network

for a parameter setting is smaller than the size of the simulated network, we removed this parameter setting. The ranges of $\lambda$ and $\gamma$ are slightly different and vary across different datasets in order to infer similarly sized networks for different datasets. We used AUPR and F-score of top $K$ edges to select the best parameter settings. We compared the algorithms using the best and all parameter settings.

AMuSR: The Inferelator-AMuSR[28] algorithm uses sparse block-sparse regression to estimate the activities of transcription factors and infer gene regulatory networks from expression datasets. The multi-task learning approach decomposes the model coefficients matrix into a dataset-specific component using a sparse penalty and a conserved component using a block-sparse penalty to capture both conserved interactions and dataset-unique interactions. It is able to incorporate prior knowledge from multiple resources and robust to false interactions in the prior network. For our simulation setting, we applied AMuSR without TFA estimation by setting worker.set_tfa(tfa_driver = False) in the SingleCellWorkflow from Inferelator 3.0 package. To be comparable across different algorithms, AMuSR was applied on the same subsample of the three simulation datasets within a stability selection framework to estimate the confidence score for each edge in the AMuSR networks. The AMuSR algorithm has two sparsity parameters: $\lambda_s$ that controls the sparsity of the network for each dataset, the block-sparse penalty $\lambda_b$ that controls the sparsity of the conserved network across all datasets. AMuSR has its own parameter selection framework (see ref. 28 for details) and uses extended Bayesian information criterion (EBIC) to select the optimal $(\lambda_s, \lambda_b)$. We additionally externally tuned the parameters by setting $c$ to {0.01, 0.02154435, 0.04641589, 0.1, 0.21544347, 0.46415888, 1, 2.15443469, 4.64158883, 10} and set $\lambda_b = c^* \sqrt{\frac{d^* log(p)}{n}}$ as suggested in the paper, where $d$ is the number of cell types, $n$ is the number of samples and $p$ is the number of genes. However, by setting $\lambda_b$ to 0 and $\lambda_s$ to 0 (the lowest sparsity settings), we found that the inferred networks are too sparse with 7–100 edges for data 1, and 71–129 edges for data 2. We kept two settings for AMuSR, one using our criteria to select the best setting based on AUPR and F-scores among different $c$ settings (AMuSR_tuned) and another version using AMuSR's default optimal parameter selection (AMuSR_default). We computed AUPR and F-score of top $K$ edges (where $K$ is the number of edges in the simulated network) for AMuSR inferred networks with optimal parameter settings for comparison with other algorithms. We compared the algorithms using the optimal and all parameter settings.

INDEP: The INDEP algorithm is the single-task framework of scMTNI which does not have the prior for sharing information across cell types and infers a regulatory network for each cell type independently. Similar to scMTNI, it models each network using a dependency network. INDEP learns the graphs for each cell type using a greedy graph learning algorithm with a score-based search, where the score contains only the data likelihood. At each iteration, the algorithm computes the change in data likelihood score[22] for all candidate regulators for each target gene, selects the best regulator for the target gene and adds this (regulator, target) edge to the current graph. INDEP has two parameters in the model: a sparsity penalty $\beta_0$ that controls penalty for adding edges, and a penalty $\beta_1$ that controls the strength of motif prior. In the simulation case, $\beta_0$ were set to {−0.005, −0.01, −0.05, −0.1, −0.5, −1}, and $\beta_1$ were set to 0. AUPR and F-score of top $K$ edges were used to select the best parameter settings (Supplementary Fig. 40). If the size of the predicted network for a parameter setting is smaller than the size of the simulated network, we removed this parameter setting. As mentioned above, we compared INDEP to other algorithms using best and all parameter settings for a dataset.

LASSO: The LASSO method uses linear regression with $L_1$ regularization. For each gene, we use the expression profiles of candidate regulators to predict the expression profiles of this gene. The regulators with non-zero coefficients are inferred as the regulators for this

**Table 2 | The statistics of the real datasets and the size of the prior networks in mouse cellular reprogramming data, human hematopoietic data from Buenrostro et al., and human fetal hematopoiesis data from Ranzoni et al.**

| Dataset | Real dataset | | Prior network | | |
|---|---|---|---|---|---|
| | # regulators | # genes | avg. # of regulators | avg. # of genes | avg. # of edges |
| Cellular reprogramming | 2036 | 12216 | 397 | 11290 | 892666 |
| Adult hematopoiesis | 1999 | 11994 | 324 | 10283 | 665931 |
| Fetal hematopoiesis (fine tree) | 2195 | 16737 | 255 | 9403 | 541813 |
| Fetal hematopoiesis (coarse tree) | 2227 | 17425 | 328 | 12308 | 865983 |

The averages are computed across the cell clusters or cell types for each dataset (cellular reprogramming data: $n = 7$, adult hematopoiesis data: $n = 8$, fetal hematopoiesis (fine tree): $n = 11$, fetal hematopoiesis (coarse tree): $n = 4$).

gene and these edges are added to the gene regulatory network. We used MATLAB implementation of LASSO regression. Similar to scMTNI, GNAT, INDEP, Ontogenet, AMuSR, LASSO was run on the same subsample of the three simulation datasets within a stability selection framework to estimate the confidence score for each edge in the networks. LASSO has only the $L_1$ penalty $\lambda$ that controls the sparsity of the network. In the simulation case, $\lambda$ were set to {0.01, 0.02, 0.03, 0.04, 0.05, 0.06}. AUPR and F-score of top $K$ edges were used to select the best parameter settings (Supplementary Fig. 41). If the size of the predicted network for a parameter setting is smaller than the size of the simulated network, we removed this parameter setting. We compared LASSO to other algorithms using the best and all parameter settings.

SCENIC: The SCENIC[30] algorithm uses GENIE3 or GRNBoost2 to infer TF-target relationships available as part of the Arboreto framework[79]. We used the GRNBoost2 algorithm with default parameters for network inference. SCENIC is based on an ensemble of trees with its own bootstrapping and hence was directly applied to each cell type-specific dataset in the simulation. SCENIC uses the feature importance score of each edge to rank the edges in the inferred network. We computed AUPR and F-score of top $K$ edges (where $K$ is the number of edges in the simulated network) for SCENIC inferred networks for comparison with other algorithms.

**Application of network inference algorithms to cellular reprogramming data**

We applied scMTNI, scMTNI+Prior, INDEP, INDEP+Prior, SCENIC, and CellOracle to the cellular reprogramming data, which contains 12,216 genes and 2036 potential regulators (Table 2). All of these methods require list of regulators and target genes information provided as input, and the same information is provided to all methods under comparison. The CellOracle algorithm is a new method that can integrate scRNA-seq profiles with non-transcriptomic data (such as bulk ATAC-seq and scATAC-seq profiles) to infer cell type-specific GRNs[21]. The algorithm is based on a regularized linear regression model and implemented in a Bayesian Ridge or Bagging Ridge framework to improve stability and reproducibility. CellOracle uses scATAC-seq data or bulk ATAC-seq data to identify accessible promoters and enhancers, and then scans TF motifs to construct a context-independent "base GRN". Subsequently, for each context, CellOracle assigns edge weights to the edges of the base GRN with the help of the context-specific scRNA-seq profiles. To infer the edge weights, CellOracle builds a regularized linear regression model to predict the expression of target gene using expression of candidate regulators. The inferred GRNs are context-specific weighted directed graphs with regression coefficients corresponding to the strength of the connections.

scMTNI and INDEP algorithms were applied within a stability selection framework to estimate edge confidence. In the stability selection framework, we subsampled the data 50 times, each with 12,216 genes and $\frac{2}{3}$ of the cells, applied the algorithms to each

subsample and used the inferred networks to estimate the confidence score for each TF-target edge in the predicted networks. In both scMTNI and scMTNI+Prior, we used the following hyper-parameter settings for the lineage structure prior $p_r = 0.2$, $p_g^{(d)} = 0.2$ and $p_m^{(d)} = 0.8$. For the sparsity prior we set $\beta_0 = -0.9$ for scMTNI, and $\beta_0 \in \{-0.9, -2, -3, -4\}$ for scMTNI+Prior. To generate the prior network, we used the matched scATAC-seq clusters to obtain TF-target prior interactions for each scRNA-seq cluster. For scMTNI+Prior which uses the scATAC-seq prior, we set $\beta_1 \in \{2, 4\}$. INDEP and INDEP+Prior were applied on the same subsampled data followed by edge confidence estimation. We used the same settings for $\beta_0$ and $\beta_1$ for INDEP as scMTNI. Final results of scMTNI+Prior used $\beta_0 = -4$ and $\beta_1 = 4$, which was determined by the distribution of edges at different confidences. Final results for INDEP+Prior used $\beta_0 = -4$ and $\beta_1 = 4$. scMTNI and INDEP were run in parallel by splitting the target gene set into subsets, e.g., of 50 genes while keeping the regulator list and other settings the same. The inferred networks of each subset target genes were concatenated as the final inferred network. The average runtime and memory usage of scMTNI and scMTNI+Prior for this dataset are reported in Supplementary Table 2. SCENIC has its own subsampling framework which can estimate an edge importance, and was applied to the entire dataset with default parameter settings. CellOracle was applied using the Bagging Ridge regression model, which has its own bootstrapping to estimate edge importance. CellOracle was applied to the entire dataset with default parameter settings and the same prior networks as for INDEP+Prior and scMTNI+Prior to enable a fair comparison of their GRN inference capabilities.

**Application of network inference algorithms to human adult hematopoietic differentiation data**

We used a similar workflow for the human hematopoietic differentiation dataset as the reprogramming system. This dataset had 11,994 genes and 1999 potential regulators (Table 2). We subsampled the scRNA-seq data for each cell cluster 50 times, each with 11,994 genes and $\frac{2}{3}$ of the cells, and applied scMTNI, scMTNI+Prior, INDEP, INDEP+Prior on each subsample to estimate the edge confidence of the GRNs. For scMTNI and scMTNI+Prior, the lineage structure prior parameters were set as follows: $p_r = 0.2$, $p_g^{(d)} = 0.2$, $p_m^{(d)} = 0.8$. The sparsity prior $\beta_0$ was set to $-0.9$ for scMTNI. For scMTNI+Prior, the sparsity prior was set $\beta_0 \in \{-0.9, -2, -3, -4\}$ and $\beta_1 \in \{2, 4\}$. For INDEP and INDEP+Prior, we used the same settings for $\beta_0$ and $\beta_1$ as scMTNI and scMTNI+Prior respectively. Final results of scMTNI+Prior are with $\beta_0 = -4$ and $\beta_1 = 4$ and final results for INDEP+Prior are using $\beta_0 = -4$ and $\beta_1 = 4$. The runtime and memory usage of scMTNI and scMTNI+Prior for this dataset are reported Supplementary Table 2. SCENIC was applied to the entire dataset with default parameter settings. CellOracle was applied to the entire dataset with default parameter settings using the same prior networks as for scMTNI+Prior and INDEP+Prior. The same list of regulators and target genes are provided to all methods under comparison.

## Application of network inference algorithms to human fetal hematopoiesis data

We applied scMTNI, scMTNI+Prior, INDEP, INDEP+Prior and CellOracle to the fine-grained lineage version of this dataset using a similar workflow as the other datasets. We applied scMTNI+Prior and CellOracle to this dataset when using the coarse lineage structure. For the fine-grained lineage, there are 16,737 genes and 2195 potential regulators. For the coarse lineage, there are 17,425 genes and 2227 potential regulators (Table 2). We subsampled the scRNA-seq data for each cell cluster 50 times, each with all genes and $\frac{2}{3}$ of the cells, and applied scMTNI, scMTNI+Prior, INDEP, INDEP+Prior on each subsample to estimate the edge confidence of the GRNs. For scMTNI and scMTNI+Prior, the lineage structure prior parameters were set as follows: $p_r = 0.2$, $p_g^{(d)} = 0.2$, $p_m^{(d)} = 0.8$. The sparsity prior $\beta_0$ was set to −0.9 for scMTNI. Final results of scMTNI+Prior are with $\beta_0 = -4$ and $\beta_1 = 4$ and final results for INDEP+Prior are using $\beta_0 = -4$ and $\beta_1 = 4$. INDEP and INDEP+Prior used the same settings for $\beta_0$ and $\beta_1$ for as scMTNI and scMTNI+Prior, respectively. The runtime performance and memory usage of scMTNI and scMTNI+Prior are reported in Supplementary Table 2. CellOracle was applied to the entire dataset with default parameter settings with the same prior networks as scMTNI+Prior and INDEP+Prior. The same list of regulators and target genes are provided to all methods under comparison.

## Evaluation

**Gold standard datasets.** To evaluate the predicted networks of different inference algorithms on real data, we downloaded and processed several gold standard datasets (Table 1). For mouse reprogramming study, we curated multiple experimentally derived networks of regulatory interactions from the literature and existing databases. The statistics of the gold standard datasets are provided in Table 1. One of these datasets is ChIP-chip or ChIP-seq based gold standard (referred to as "ChIP") from ESCAPE (http://www.maayanlab.net/ESCAPE/) or ENCODE databases[34,35] (https://www.encodeproject.org/), which contains ChIP-chip or ChIP-seq experiments in mouse ESCs. The second dataset is a knock down-based gold standard (referred to as "Perturb"), which is derived from regulator perturbation followed by global transcriptome profiling[34,36]. We took a union of the networks from LOGOF (loss or gain of function) based gold standard networks from the ESCAPE database[34] and the networks from Nishiyama et al.[36] as the perturbation interactions. Finally, we took the intersection of the interactions between ChIP and knock-down based gold standards to create the third gold standard network referred to as "ChIP+Perturb".

For human hematopoietic cell types, we have five gold standard datasets. Two gold standard datasets were a ChIP-based (Cus_ChIP) and a regulator knock down-based (Cus_KO) dataset in GM12878 lymphoblastoid cell line downloaded from Cusanovich et al.[47]. For the knock down dataset, we had TF-target relationships at two $p$-value thresholds, 0.01 and 0.05. We used the TF-target relationships at 0.01 to have a more stringent gold standard. The third gold standard was from human hematopoietic cell types from the UniBind database (https://unibind.uio.no/)[46], which has high confidence TF binding site predictions from ChIP-seq experiments. To obtain the TF-gene network, we mapped TF binding sites to the nearest gene if there is overlap between the TF binding sites and the promoter of the gene defined by ±5000 bp of the gene TSS. If multiple ChIP-seq datasets were available for the same TF in a given cell type, we took the union of TF-gene edges for the same cell type. We took the union of these individual cell type-specific gold standards to create our Unibind gold standard (UniBind). Finally, we took the intersection of the ChIP-based gold standards with the knock down based gold standards, to produce the fourth and fifth gold standards, Unibind+Cus_KO and CusChIP+Cus_KO. The statistics of the gold standard datasets are provided in Table 1.

**Area under the precision recall curve.** To evaluate the performance of scMTNI and other algorithms, we compared the inferred networks to the simulated networks or interactions from the gold standard datasets based on Area under the precision recall curve (AUPR). Edge weights for all but the SCENIC and CellOracle algorithms were obtained using stability selection. Both SCENIC and CellOracle have internal bootstrapping or bagging approaches to estimate confidence in the inferred edges. In our stability selection framework, we generated $N$ random subsamples of the data, inferred a network for each subsample, and calculated a confidence score for each edge as the fraction of how many times this edge was present in the inferred networks across all subsamples. Next, we ranked the edges by the confidence score and estimated precision and recall as a function of edge confidence. Precision $P$ is defined as the fraction of the number of edges that are true positives among the total number of predicted edges. Recall $R$ is defined as the fraction of the number of edges that are true positives among the total number of true edges. Then, we plotted the precision recall curve and estimated the area under this curve using the AUCCalculator package developed by Davis et al.[80]. The area under the precision recall curve is computed as an overall assessment of the inferred networks compared to "true" networks. The higher AUPR, the better the performance. For the real scRNA-seq datasets, we filtered the inferred networks to include TFs and targets that were in the gold standard.

**F-score.** While AUPR uses a ranking of the edges, F-score is a metric to compare a set of predicted edges to a set of "true" edges. F-score is defined as the harmonic mean of the precision (P) and recall (R), $F - score = \frac{2*P*R}{P+R}$. F-score enables us to control for the number of edges across network inference algorithms as these can vary significantly across algorithms. To control for number of edges in the predicted networks, we ranked the predicted network by the confidence score or edge weight, selected top $K$ edges and computed F-score compared to simulated networks or gold standard networks. $K$ in the simulated datasets corresponded to the size of the simulated networks. For the real datasets, we considered top 500, 1000, 2000 edges. We obtained the top $K$ edges after filtering the inferred networks based on the TFs and targets in the gold standard networks. The higher the F-score, the better the performance.

**Predictable transcription factors (TFs).** Predictable TFs was defined based on the gold standard datasets similar to McCalla et al.[18]. For each TF's target set in the gold standard network, we computed its overlap with the predicted targets in the inferred network and used the hypergeometric test to assess the significance of overlap. We consider a TF to be predictable if the $P$-value < 0.05. We count the total number of predictable TFs for each algorithm as a metric of evaluation. The higher the number of predictable TFs, the better the performance.

## Examining network dynamics on cell lineages

We used several global and subnetwork-level methods to examine how regulatory networks change on a cell lineage. These include F-score based comparison of all pairs of networks on the lineage, k-means based edge clustering and Latent Dirichlet Allocation (LDA) model.

**F-score based analysis of inferred network change along cell lineage tree.** To examine the overall conservation and divergence between the inferred cell type-specific networks along the cell lineage tree, we computed F-score on the predicted networks between each pair of cell types and applied hierarchical clustering on the inferred networks based on the F-score. To compute F-score, we selected top $X$ edges ranked by confidence score to obtain a reliable network for each cell type. This was 4k in the mouse reprogramming dataset and 5k for the hematopoietic differentiation datasets. We visualized the

dendrogram obtained from the hierarchical clustering and compared this to the original cell lineage tree.

**k-means based edge clustering.** For each cell cluster, we selected top $K$ edges, where $K$ was close to the median number of edges with at least 80% confidence across all cell types. This corresponded to 4k edges for the mouse reprogramming dataset, 5k edges for the hematopoietic differentiation dataset from Buenrostro et al. and 5k and 1k edges for the coarse and fine-grained lineage structure of the fetal hematopoiesis dataset from Ranzoni et al. We merged the confidence score of each edge across all cell types as an edge by cell type matrix, each entry corresponding to the edge confidence with as many edges as in the union of top $K$ edges from any cell type. We applied k-means clustering on this matrix to find subnetworks with different patterns of conservation. We examined a range of number of clusters from $k = 5$ to 30 and selected the smallest $k$ at which silhouette coefficient was high.

**Latent Dirichlet Allocation (LDA) model for regulatory network rewiring.** We adopted Latent Dirichlet Allocation (LDA) to examine subnetwork level rewiring as described in TopicNet[40]. LDA was originally developed to cluster documents based on their word distributions. Each document, $i$ is assumed to have a certain composition of topics, as captured by a $\theta_i$ parameter and each topic, $k$, is assumed to have a specific distribution of words denoted by a $\varphi_k$ parameter. In the application of LDA to a regulatory network, we first concatenated the TF by target network across cell types to have as many rows as there are TFs times the number of cell types. Each TF in a cell type is treated as a document and its targets are treated as words in the document. The topic distribution for all documents constitutes a $M \times K$ matrix for document-topic distribution, where $M$ is the total number of TFs in any of the networks and $K$ is the total number of topics. The distribution of words (genes) in each topic is captured by a $K \times V$ matrix for $V$ genes. Each gene can be assigned to a topic based on its maximum probability across topics. We applied LDA to the 80% confidence networks of all cell clusters or types inferred from scMTNI+Prior with 10 or 15 topics and found 10 topics to be suitable for all three datasets. We extracted the subnetworks in each cell type associated with each topic by obtaining the induced graph for the genes and regulators associated with each topic and visualized the giant components of each network to identify change across cell clusters within the same topic. To interpret the topics in each cell type, we tested the genes in the cell type-specific subnetwork for each topic for enrichment of gene ontology (GO)[81] processes using a hypergeometric test with FDR correction. We define the gene set for each topic to include the cell-type specific regulators and targets per cell type. We used an FDR < 0.01 to determine significant enrichment (Supplementary Figs. 11, 21, 29). These results are described in Supplementary Figs. 8-10 for mouse cellular reprogramming, in Supplementary Figs. 18–20 for the hematopoietic differentiation data from Buenrostro et al., in Supplementary Figs. 25–28 for the fetal hematopoiesis fine-grained lineage and in Supplementary Figs. 31–35 for the fetal hematopoiesis coarse lineage data.

**Statistics and reproducibility.** In the scATAC-seq reprogramming experiment, six samples representing different time points of the reprogramming study were used. The sample size is the number of biological samples. We chose six samples to analyze because these specific timepoints, along with MEFs and ESCs, provide sufficient coverage on the various states and progression of cells during the reprogramming process. One biological replicate for each sample data was used for analysis. Previous experiments were conducted in which cells were reprogrammed using identical conditions and reagents (see Tran et al.[32]). The setup of experiments in this paper assume that one experimental replicate and one scATAC-seq submission for each sample reflects the same reprogramming time course observed in our previous experiments. For randomization, MEFs from a single embryo

were randomly seeded at a density of 5000 cells per well in 6-well plates. Blinding was not applicable to this study as no portion of this data can be skewed based on participant's knowledge of the experiment. All cells from the reprogramming plates were collected during scATAC-seq submission and the scATAC library prep and sequencing portions were performed by unbiased third parties who have no knowledge of any experimental details.

Network inference was done in a stability selection mode where we drew multiple subsamples from the original data. Each subsample's size was set to 2/3 of the number of cells in the dataset. This number was determined to enable sufficient number of cells for each subsample. Subsamples were generated by selecting uniformly at random samples from our full dataset. We have provided code, scripts, inputs and outputs from our experiments to enable replication of our study. For data exclusion, cells with low read depth and genes with fewer than 5 or 20 measurements were filtered from downstream analysis. Some cell clusters were excluded if they had either no or too few scRNA-seq cells. Cluster C1 for the hematopoietic differentiation data from Buenrostro et al. was removed from evaluation using the gold standards due to very few TFs overlapping the gold standards compared to the other cell clusters.

### Reporting summary

Further information on research design is available in the Nature Portfolio Reporting Summary linked to this article.

### Data availability

The reprogramming scATAC-seq dataset generated in this study has been deposited to Gene Expression Omnibus (GEO) with accession ID GSE208620. The scRNA-seq datasets for the same time points from Tran et al.[32] were downloaded from Gene Expression Omnibus (GEO) with accession ID GSE108222. The processed cluster-specific scRNA-seq matrices and the prior networks for reprogramming study are available at Zenodo https://zenodo.org/record/7879228[82].

The scRNA-seq data for human hematopoietic differentiation from Buenrostro et al. were downloaded from Data S2 of Buenrostro et al. (https://ars.els-cdn.com/content/image/1-s2.0-S009286741830446X-mmc4.zip) and the scATAC-seq data were downloaded from Chen et al.[72] (https://github.com/pinellolab/scATAC-benchmarking/tree/master/Real_Data/Buenrostro_2018). The scATAC-seq data are also available from GEO accession GSE96772. The scRNA-seq data (Data S2 from Buenrostro et al.,) and the scATAC-seq data have been additionally uploaded to Zenodo https://zenodo.org/record/7879228. The processed datasets for human hematopoietic differentiation are available at Zenodo https://zenodo.org/record/7879228.

The scRNA-seq (gene by cell) and scATAC-seq (peak by cell) data matrices for the human fetal hematopoietic differentiation data from Ranzoni et al. were obtained from https://gitlab.com/cvejic-group/integrative-scrna-scatac-human-foetal. These are also available at ArrayExpress: E-MTAB-9067 for scRNA-seq and E-MTAB-9068 for scATAC-seq. The cluster-specific scRNA-seq matrices and the prior networks are available at Zenodo https://zenodo.org/record/7879228.

For the mouse reprogramming study, the ChIP-based gold standard datasets were downloaded from ESCAPE (http://www.maayanlab.net/ESCAPE/) and ENCODE databases[34,35] (https://www.encodeproject.org/). The Perturbation-based gold standard networks were constructed from a union of the networks from LOGOF (loss or gain of function) based gold standard networks from ESCAPE database[34] and the networks from Nishiyama et al.[36]. The mouse gold standard datasets are available at Zenodo https://zenodo.org/record/7879228.

For the human hematopoietic data, two gold standard datasets were a ChIP-based (Cus_ChIP) and a regulator knock down-based (Cus_KO) dataset in GM12878 lymphoblastoid cell line downloaded from Cusanovich et al.[47]. The third gold standard from ChIP-seq experiments in human hematopoietic cell types was downloaded from

the UniBind database (https://unibind.uio.no/)[46]. The human gold standard datasets are available at Zenodo https://zenodo.org/record/7879228.

The source data underlying Figs. 2–8, Supplementary Figs. 2, 3, 5, 7–10, 12, 14, 15, 17–20, 22, 24–28, 20–29, 30–49, the cluster-specific scRNA-seq matrices and the prior networks for all datasets and scMTNI inferred consensus networks are available at Zenodo https://zenodo.org/record/7879228[82]. All other relevant data supporting the key findings of this study are available within the article and its Supplementary Information files or from the corresponding author upon reasonable request. Source data are provided with this paper.

## Code availability

The scMTNI code and custom scripts to process data and compute various validation metrics and perform dynamic network analysis are available at https://github.com/Roy-lab/scMTNI and Zenodo https://doi.org/10.5281/zenodo.7854535[83]. Custom scripts include shell scripts, python scripts, R scripts and MATLAB scripts and we used R version 3.5.1, MATLAB version R2014b, and Python version 3.6.12 to perform data analysis. The scATAC-seq data was processed through CellRanger ATAC pipeline (Version 1.1.0). The simplified implementation of the pagoda pipeline for normalizing scRNA-seq data for depth and variance stabilization is available at https://github.com/Roy-lab/scMTNI/blob/master/Scripts/Integration/adjustVariance_depth_Generic.R. R package rliger version 1.0.0 was used to integrate scRNA-seq and scATAC-seq data, and the R script is available at https://github.com/Roy-lab/scMTNI/tree/master/Scripts/Integration/. To generate prior networks, we used MACS v2.1.0 to call ATAC-seq peaks to generate prior networks and used custom code for mapping TF binding peaks to genes, which is available at https://github.com/Roy-lab/scMTNI/tree/master/Scripts/genPriorNetwork/. The scripts for evaluation based on AUPR and F-score are available at https://github.com/Roy-lab/scMTNI/tree/master/Evaluation/. The scripts for dynamic network analysis are available at https://github.com/Roy-lab/scMTNI/tree/master/Scripts/Network_Analysis/.

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

## Acknowledgements

We thank the Center for High Throughput Computing at University of Wisconsin-Madison for computational resources. This work is supported by the National Institutes of Health NIGMS grant 1R01GM117339 (S.R., S.Z., A.F.S.) and 1R01GM144708-01A1 (S.Z., S.G.M, S.R., S.H.), the Department of Energy grant DE-SC0021052 (S.Py.), and grant 2R01GM113033 (R.S., S.Pi.). We thank Dr. Jason Buenrostro for help with accessing the scRNA-seq data for the adult hematopoiesis dataset from Buenrostro et al.

## Author contributions

S.Z. and S.R. designed the scMTNI algorithm and experiments. S.Z. implemented the code and performed most of the experiments. S.Py. contributed towards creation of the gold standards and evaluating selected algorithms. S.G.M. and S.H. contributed toward evaluation of algorithms on the fetal hematopoiesis dataset. S.Pi. and R.S. generated the scATAC-seq data for the reprogramming experiments. A.F.S. contributed towards processing the scATAC-seq data for the reprogramming experiments and sequence-specific motifs from the Cis-BP database and assisted with collection of gold standards from the hESC and mESC cell lines. All authors contributed toward writing the manuscript.

## Competing interests

The authors declare no competing interests.
