## [Peer Review File · Nature Communications]

REVIEWER COMMENTS

Reviewer #1 (Remarks to the Author):

In the submitted draft manuscript, the authors propose a method to integrate scRNA-seq and scATAC-seq measurements and to model network dynamics on a cell lineage. The manuscript has four main parts

1. method establishment.
2. Benchmarking on simulated data
3. Previously published data - haematopoietic differentiation
4. new scATACseq data + published scRNAseq data - MEFToiPS reprogramming.

Part 1 and 2 of the manuscript are good and part 3 and 4 are entirely unpublishable in current form in my view. I will provide reasons below.

Part 1 and 2: The method uses cell lineage tree as an input to infer regulatory networks from ATAC-seq and scRNA-seq data for each cell cluster. The framework of the method is good and with the simulated data, the authors make a case that the method works. The biggest criticism for this part is that the authors do not make it clear what is the biggest improvement to state-of-art? i.e. is there method particularly fast, user-friendly or much more reliable ??

Part 3 and 4: These two parts of the manuscript unfortunately fail to make a case for the method. Firstly there are major biological flaws on both biological system analyses. The biggest off-course is a fact that the authors use linear differentiation (HSC) or dedifferentiation (iPS) data and build a lineage tree with it. Given that there is no real tree structure in the input data, the analysis and interpretation remains really superficial. e.g. what is the identity of cluster 2 in iPS data? there is one line in the manuscript that it is stalled dedifferentiation but obviously not much support or evidence for that claim. Given that a signal for that cluster comes from day3 scATACseq and not even supported by RNA-seq, my assumption would be that is it an artefact. Similarly for the HSC dataset, GMP and CMP are intermediate progenitor populations so again there is no real tree structure in the data. In both cases, I fail to see what more have the authors learned compared to the regulatory network one would obtain by learning in on HSC, monocyte, MEF and iPS cells?

In short, the authors have come up with a really nice concept of using cell lineage tree for regulatory network inference and the benchmarking on simulated data. Unfortunately, the authors completely fail to make a case for their method in real biological systems, partly because of wrong choice of biological systems. The biggest strength of the method is the information from lineage tree and the two biological systems used have no inherent tree structure but are rather linear differentiation or de-differentiation structure.

Reviewer #2 (Remarks to the Author):

The authors presented scMTNI, which is a multi-task learning framework that can learn cell type specific gene regulatory networks (GRNs) from scATAC-seq and scRNA-seq data. The method uses relationships between GRNs of different cell types and prior graphs obtained from the scATAC-seq data as prior knowledge, and integrates this prior information into the probabilistic framework used to infer the GRNs. The use of the prior graphs in this work is novel and the analysis on results of biological datasets are insightful. The manuscript can be further improved through addressing the following points:

1. The authors have compared their methods, scMTNI and scMTNI+prior with other single-task or multi-task GRN inference methods, and the results showed the advantages of multi-task methods.

scMTNI has not been compared with other methods that use both chromatin accessibility and gene expression data to infer GRN, though such methods are rare. CellOracle (<https://www.biorxiv.org/content/10.1101/2020.02.17.947416v3>) is one such method that I suggest the authors to compare their methods with.

2. From the optimization procedure used in scMTNI, this method can potentially be computationally expensive. Can the authors please provide running time analysis of scMTNI and scMTNI+prior? Also, on simulated datasets, the GRN network size is relatively small (15 regulators and 65 genes). What are the network sizes inferred on the real datasets?

3. Do scMTNI and baseline methods require TF and target gene information (that is, which nodes in the GRN are TFs and which are target genes) as input? In other words, is this information provided to all methods under comparison?

4. From how the method works, it seems that the methods do not necessarily need single cell ATAC-seq data – it can also work with bulk ATAC-seq data if the bulk samples correspond to the cell types under consideration. If this is the case, it would be helpful to mention this in the paper to broaden the applicability of the methods.

5. Some points regarding the figures: (1) Fig. 2B-C: should y-axis labels be “cell type 1”, “cell type 2”, ..., instead of “cell1”, “cell2”, ...? (2) Fig. 3B: this plot is hard to read and what information is expected from this plot is not clear. From the integration perspective, we would expect that cells from atac and rna labeled with the same Day are mixed. Maybe one can use multiple supplementary plots, each showing only cells from one Day (or cell type).

Point by point response to Reviewer Comments.

Reviewers' comments are in black font color. Our responses are in blue font color.

Reviewer #1 (Remarks to the Author):

In the submitted draft manuscript, the authors propose a method to integrate scRNA-seq and scATAC-seq measurements and to model network dynamics on a cell lineage. The manuscript has four main parts

1. method establishment.
2. Benchmarking on simulated data
3. Previously published data - haematopoietic differentiation
4. new scATACseq data + published scRNAseq data - MEFtoIPS reprogramming.

Part 1 and 2 of the manuscript are good and part 3 and 4 are entirely unpublishable in current form in my view. I will provide reasons below.

We thank the reviewer for this comment. Below we provide a point by point response to address the concerns raised by the reviewer.

Part 1 and 2: The method uses cell lineage tree as an input to infer regulatory networks from ATAC-seq and scRNA-seq data for each cell cluster. The framework of the method is good and with the simulated data, the authors make a case that the method works. The biggest criticism for this part is that the authors do not make it clear what is the biggest improvement to state-of-art? i.e. is there method particularly fast, user-friendly or much more reliable ??

We thank the reviewer for this summary and this comment. The biggest improvement of scMTNI is that it is more reliable and flexible, both for incorporating different topologies of cell lineages, for incorporating different sources of accessibility as priors and for including regulators that might not have motif support. The cell lineage structure can be linear or branching and can be obtained based on literature or computational inference from scRNA-seq data. The lineage modeling is accomplished with scMTNI's multi-task learning framework and was important to examine network dynamics and identify regulators and GRN components associated with different parts of the lineage tree. We have updated the **Abstract**, **Introduction** and **Discussion** to emphasize this point.

Part 3 and 4: These two parts of the manuscript unfortunately fail to make a case for the method. Firstly there are major biological flaws on both biological system analyses. The biggest off-course is a fact that the authors use linear differentiation (HSC) or dedifferentiation (iPS) data and build a lineage tree with it. Given that there is no real tree structure in the input data, the analysis and interpretation remains really superficial. E.g. what is the identity of cluster 2 in iPS data? There is one line in the manuscript that it is stalled dedifferentiation but obviously not much support or evidence for that claim. Given that a signal for that cluster comes from day3 scATACseq and not even supported by RNA-seq, my assumption would be that is it an artefact. Similarly for the HSC dataset, GMP and CMP are intermediate progenitor populations so again there is no real tree structure in the data. In both cases, I fail to see what more have the authors learned compared to the regulatory network one would obtain by learning in on HSC, monocyte, MEF and iPS cells?

We thank the reviewer for this comment. We note that scMTNI does not require the trajectory to represent a branching topology, it can also be applied to a linear topology. Our rationale for first computationally inferring a tree structure from the data was the heterogeneity in the two systems we studied, reprogramming and human hematopoiesis. For the reprogramming system, although the cells were collected at distinct time points, the reprogramming process is heterogeneous and cells from each timepoint can be mixture of different stages of the reprogramming system. Furthermore, Tran et al. ^[1] had observed a bifurcating trajectory using Monocle on the scRNA-seq data (for the A2S condition), which corresponded to cell populations that reprogrammed while those that stalled. To further support our trajectory structure, we applied Monocle to our current scRNA-seq dataset used for scMTNI (FBS condition). As in the A2S condition, Monocle on the scRNA-seq data under FBS condition outputs a branching trajectory where a fraction of the cells reach the pluripotent state (State 7, **Supplementary Figure 4, reproduced below as Reviewer Fig 1A**), however a number of cells exit this trajectory and go to a different state (State 4 and 6, **Reviewer Figure 1A**). Cells in these states represent a mixture of cells from the four time points of D3, D6, D9 and D12. We compared the proportion of cells in our clusters with the cells assigned to each state and find that our cell clusters C7, C3 and C2 have the highest overlap with these states (C7, C3, C2 overlap State 4 and C7 and C3 overlap State 6, **Reviewer Figure 1B**). This analysis provides further support for the large amount of heterogeneity in this system and that the time point labels of the cells is not a biologically meaningful partition of the cells other than for MEF and ESC.

We have also now expanded on the characterization of the cells in the C2, C3 and C7 clusters using the scMTNI networks. In particular, using our dynamic network analysis, we identified which regulators exhibit a relative gain or loss of edges in cell clusters on the C7-C2-C3 branch compared to the C1-C5 branch. Our analysis identified MEF-specific regulators (e.g. Aebp1, Nme2, Loxl2) to have more regulatory edges in this branch, and a corresponding loss of cell cycle and stem cell maintenance regulatory connections (e.g. Top2a and Esrrb). We have expanded the results section, **“scMTNI predicts key regulatory nodes and GRN components that are rewired during reprogramming”** to better emphasize these results and also included our Monocle analysis in **Supplementary Figure 4**.

Our rationale for using a computationally inferred tree for the adult hematopoiesis dataset from Buenrostro et al. is similar and motivated by the observation of the original authors that there is a large amount of heterogeneity in the progenitor CMP and GMP populations and the precise ordering and grouping of cells is not immediately clear. Here again, we used our dynamic network analysis of scMTNI inferred networks to characterize the cell populations that had more CMP proportion (C7, C10) compared to GMP (C6, C9) and we find evidence of an Erythroid lineage with regulators KLF1, FLI1 and PBX1 to have more connections in the CMP cell clusters compared to the GMP cell clusters. Further between C7 and C10, we see the connectivity of several of these regulators to be higher in the C7 cluster compared to C10, indicative of additional fine-grained differences in these cell clusters. In contrast, the GMP cell clusters exhibit greater connections associated with immune system regulators such as IRF8 which suggests the presence of a lymphoid lineage. Furthermore, between C6 and C9, we observed more pronounced hubs for IRF8 in C6 compared to C9. We have expanded our results section, **“Inferring shared and lineage-specific regulators for hematopoietic differentiation”** to highlight these observations.

In addition to expanded analysis of the two datasets in our original manuscript, we have now included a new dataset for fetal hematopoiesis differentiation from the Hematopoietic stem cell state to different lymphoid, myeloid lineages from Ranzoni et al ^[2]. We used the known branching lineage structure for these cell populations as input to scMTNI. In addition to identifying known and novel regulators associated with different lineage decisions, our analysis also demonstrated superior performance of scMTNI compared to existing methods such as CellOracle ^[3]. We have included these results in two new

Result sections (“Inferring gene regulatory networks in human fetal hematopoiesis”, “Examining dynamics of GRN components for fetal hematopoiesis”), two new main figures (Figures 7 and 8), and several Supplementary Figures (Supplementary Figures 22-35).

In short, the authors have come up with a really nice concept of using cell lineage tree for regulatory network inference and the benchmarking on simulated data. Unfortunately, the authors completely fail to make a case for their method in real biological systems, partly because of wrong choice of biological systems. The biggest strength of the method is the information from lineage tree and the two biological

systems used have no inherent tree structure but are rather linear differentiation or de-differentiation structure.

We thank the reviewer for this comment. We believe that our new analysis and experiments including the addition of the fetal hematopoiesis dataset and clarifications are able to demonstrate the benefits of scMTNI to infer and analyze GRN dynamics on known and inferred linear and branching topologies.

Reviewer #2 (Remarks to the Author):

The authors presented scMTNI, which is a multi-task learning framework that can learn cell type specific gene regulatory networks (GRNs) from scATAC-seq and scRNA-seq data. The method uses relationships between GRNs of different cell types and prior graphs obtained from the scATAC-seq data as prior knowledge, and integrates this prior information into the probabilistic framework used to infer the GRNs. The use of the prior graphs in this work is novel and the analysis on results of biological datasets are insightful.

We thank the reviewer for their comments.

The manuscript can be further improved through addressing the following points:

1. The authors have compared their methods, scMTNI and scMTNI+prior with other single-task or multi-task GRN inference methods, and the results showed the advantages of multi-task methods. scMTNI has not been compared with other methods that use both chromatin accessibility and gene expression data to infer GRN, though such methods are rare. CellOracle (<https://www.biorxiv.org/content/10.1101/2020.02.17.947416v3>) is one such method that I suggest the authors to compare their methods with.

Thanks for the suggestion. We have added the comparison of scMTNI with CellOracle for each of the original datasets. We also included a new dataset to address Reviewer 1's comments and we show scMTNI comparison against CellOracle on this dataset as well. We have updated the **Results, Methods** and **Discussions** accordingly. This experiment has resulted in updates to **Figures 3** and **5**, and new **Figure 7** and **Supplementary Figures 5-7, 12-17, 22-24 and 30**. Our results show that while CellOracle has good performance on CHIP-based gold standards, it suffers on some perturbation gold standards and as such is sensitive to the coverage of the input prior network.

2. From the optimization procedure used in scMTNI, this method can potentially be computationally expensive. Can the authors please provide running time analysis of scMTNI and scMTNI+prior? Also, on simulated datasets, the GRN network size is relatively small (15 regulators and 65 genes). What are the network sizes inferred on the real datasets?

We have provided running time analysis of scMTNI and scMTNI+prior in **Supplementary Table 2**. The average runtime for scMTNI is 0.6-0.8 hours, and 0.29-0.38 hours for scMTNI+prior for 50 genes and 1999-2195 regulators for all cell types together in real datasets.

For cellular reprogramming data, there are 2036 regulators and 12216 genes. For human hematopoietic differentiation data from Buenrostro et al, there are 1999 regulators and 11994 genes. The human fetal hematopoiesis data has 2195 regulators and 16737 genes in the fine-grained lineage data and 2227 regulators and 17425 target genes in the coarse lineage data. These statistics are included in **Table 4**.

3. Do scMTNI and baseline methods require TF and target gene information (that is, which nodes in the GRN are TFs and which are target genes) as input? In other words, is this information provided to all methods under comparison?

Yes, scMTNI and baseline methods require TF and target gene information as input. This information is provided to all methods under comparison. We note that TFs can be targets too. We have clarified this in the corresponding **Methods** sections, “**Application of network inference algorithms on simulated datasets**”, “**Application of network inference algorithms to cellular reprogramming data**”, “**Application of network inference algorithms to human adult hematopoietic differentiation data**”, “**Application of network inference algorithms to human fetal hematopoiesis data**”.

4. From how the method works, it seems that the methods do not necessarily need single cell ATAC-seq data – it can also work with bulk ATAC-seq data if the bulk samples correspond to the cell types under consideration. If this is the case, it would be helpful to mention this in the paper to broaden the applicability of the methods.

Thanks for the suggestion. Indeed, scMTNI can be used with different sources of prior information including those from bulk accessibility data or from sequence alone and can be run even without prior information. We have added this in the **Results** section: “**Single-cell Multi-Task learning Network Inference (scMTNI) for defining regulatory networks on cell lineages**” and also in the **Discussion** to emphasize the flexibility offered by scMTNI.

5. Some points regarding the figures: (1) Fig. 2B-C: should y-axis labels be “cell type 1”, “cell type 2”, ..., instead of “cell1”, “cell2”, ...? (2) Fig. 3B: this plot is hard to read and what information is expected from this plot is not clear. From the integration perspective, we would expect that cells from atac and rna labeled with the same Day are mixed. Maybe one can use multiple supplementary plots, each showing only cells from one Day (or cell type).

Thank you for these suggestions. We have updated the figures as follows:

1. We have changed the y-axis labels for **Figure 2B-C** to be “cell type 1”, “cell type 2”, etc. We have also updated corresponding **Supplementary Figures 36-41**.
2. We have made separate plots for each sample shown in **Figure 3B** in a new **Supplementary Figure 3**. Based on these plots, we can see that scRNA and scATAC labeled cells of the same Day are well integrated.

Reference:

1. Khoa A Tran, Stefan J Pietrzak, Nur Zafirah Zaidan, Alireza Fotuhi Siahpirani, Sunnie Grace McCalla, Amber S Zhou, Gopal Iyer, Sushmita Roy, and Rupa Sridharan. Defining reprogramming checkpoints from single-cell analyses of induced pluripotency. *Cell reports*, 27(6):1726-1741, 2019.
2. Anna Maria Ranzoni, Andrea Tangherloni, Ivan Berest, Simone Giovanni Riva, Brynelle Myers, Paulina M Strzelecka, Jiarui Xu, Elisa Panada, Irina Mohorianu, Judith B Zaugg, et al. Integrative single-cell rna-seq and atac-seq analysis of human developmental hematopoiesis. *Cell stem cell*, 28(3):472–487, 2021.
3. Kenji Kamimoto, Blerta Stringa, Christy M Hoffmann, Kunal Jindal, Lilianna Solnica-Krezel, and Samantha A Morris. Dissecting cell identity via network inference and in silico gene perturbation. *Nature*, pages 1–10, 2023.

REVIEWERS' COMMENTS

Reviewer #1 (Remarks to the Author):

The Authors have addressed major comments in the revision of the manuscript and I believe that this manuscript is indeed of value to the scientific community and therefore I recommend its publication.

Reviewer #2 (Remarks to the Author):

The authors have addressed my concerns in the revised manuscript.